# Shared functional specialization in transformer-based language models and the human brain

Sreejan Kumar [1,6] ✉, Theodore R. Sumers [2,6] ✉, Takateru Yamakoshi [3], Ariel Goldstein[4], Uri Hasson [1,5], Kenneth A. Norman [1,5], Thomas L. Griffiths [2,5], Robert D. Hawkins[1,5] & Samuel A. Nastase [1] ✉

When processing language, the brain is thought to deploy specialized computations to construct meaning from complex linguistic structures. Recently, artificial neural networks based on the Transformer architecture have revolutionized the field of natural language processing. Transformers integrate contextual information across words via structured circuit computations. Prior work has focused on the internal representations ("embeddings") generated by these circuits. In this paper, we instead analyze the circuit computations directly: we deconstruct these computations into the functionally-specialized "transformations" that integrate contextual information across words. Using functional MRI data acquired while participants listened to naturalistic stories, we first verify that the transformations account for considerable variance in brain activity across the cortical language network. We then demonstrate that the emergent computations performed by individual, functionally-specialized "attention heads" differentially predict brain activity in specific cortical regions. These heads fall along gradients corresponding to different layers and context lengths in a low-dimensional cortical space.

Language comprehension is a fundamentally constructive process. We resolve local dependencies among words to assemble lower-level linguistic units into higher-level units of meaning[1–6], ultimately arriving at the kind of narratives we use to understand the world[7,8]. For example, if a speaker refers to "the secret plan," we implicitly process the relationships between words in this construction to understand that "secret" modifies "plan." At a higher level, we use the context of the surrounding narrative to understand the meaning of this phrase—what does the plan entail, who is keeping it secret, and who are they keeping it secret from? This context may comprise hundreds of words unfolding over the course of several minutes. The human brain is thought to implement these processes via a series of functionally

specialized computations that transform acoustic speech signals into actionable representations of meaning[9–15].

Traditionally, neuroimaging research has used targeted experimental manipulations to isolate particular linguistic computations—for example, by manipulating the presence/absence or complexity of a given syntactic structure—and mapped these computations onto brain activity in controlled settings[15–19]. While these findings laid the groundwork for a neurobiology of language, they have limited generalizability outside the laboratory setting, and it has proven difficult to synthesize them into a holistic model that can cope with the full complexity of natural language. This has prompted the field to move toward more naturalistic comprehension paradigms[20–23]. However,

[1]Princeton Neuroscience Institute, Princeton University, Princeton, NJ 08540, USA. [2]Department of Computer Science, Princeton University, Princeton, NJ 08540, USA. [3]Faculty of Medicine, The University of Tokyo, Bunkyo-ku, Tokyo 113-0033, Japan. [4]Department of Cognitive and Brain Sciences and Business School, Hebrew University, Jerusalem 9190401, Israel. [5]Department of Psychology, Princeton University, Princeton, NJ 08540, USA. [6]These authors contributed equally: Sreejan Kumar, Theodore R. Sumers. ✉e-mail: sreejank@princeton.edu; sumers@princeton.edu; snastase@princeton.edu

these paradigms introduce their own challenges: principally, how to explicitly quantify the linguistic content and computations supporting the richness and expressivity of natural language[24–30].

In recent years, the field of natural language processing (NLP) has been revolutionized by a new generation of deep neural networks capitalizing on the Transformer architecture[31–33]. Transformers are deep neural networks that forgo recurrent connections[34,35] in favor of layered "attention head" circuits, facilitating self-supervised training on massive real-world text corpora. Following pioneering work on word embeddings[36–38], the Transformer architecture represents the meaning of words as numerical vectors in a high-dimensional "embedding" space where closely related words are located nearer to each other. However, while the previous generation of embeddings assign each word a single static (i.e., non-contextual) meaning, Transformers process long sequences of words simultaneously to assign each word a context-sensitive meaning. The core circuit motif of the Transformer—the attention head—incorporates a weighted sum of information exposed by other words, where the relative weighting "attends" more strongly to some words than others. The initial embeddings used as input to the Transformer are non-contextual. Within the Transformer, attention heads in each layer operate in parallel to update the contextual embedding, resulting in surprisingly sophisticated representations of linguistic structure[39–42].

The success of Transformers has inspired a growing body of neuroscientific work using them to model human brain activity during natural language comprehension ([43–53]; cf.[52,54]). These efforts have focused exclusively on the "embeddings"—the Transformer's representation of linguistic content—and have largely overlooked the "transformations"—the actual computations performed by the attention heads. Although no functionally-specific language modules are built into the architecture at initialization, recent work in NLP has revealed emergent functional specialization in the network after training[55,56]. That is, particular attention heads are shown to selectively implement interpretable linguistic operations. For example, attention head 10 in the eighth layer of BERT appears to be specialized for resolving the direct object of a verb (e.g., in "the boy in the yellow coat greeted his teacher", the verb "greeted" attends to "boy"), whereas head 11 in the same layer closely tracks nominal modifiers (e.g., attending to "coat" in the phrase modifying "boy"). This is in contrast to probabilistic syntactic parsers[29,54,57], which learn to reproduce a predefined set of syntactic labels to construct parse trees. The transformations do not explicitly disentangle syntax from the meaning of words and do not rely on predefined labels; instead they learn to approximate whatever contextual structures are useful for accurately predicting words in real-world text. Although the individual heads that implement these computations operate independently, in parallel, their transformations are ultimately "fused" together to form the resulting embedding. Thus, unlike the embeddings, the transformations at a given layer can be disassembled into the specialized computations performed by the constituent heads. These transformations are of particular theoretical interest, because they are the unique component of the circuit that allows information to flow between words: whatever syntactic or contextual information impacts the meaning of the current word is introduced solely via the transformations.

In the current work, we argue that the headwise transformations—the functionally specialized contextual computations implemented by individual attention heads—can provide a complementary window onto linguistic processing in the brain (Fig. 1A). A neurocomputational theory of natural language processing must ultimately specify how meaning is constructed across words. The Transformer architecture provides explicit access to a candidate mechanism for quantifying how the meaning of past words is incorporated into the meaning of the current word. If this is an important part of human language processing, these transformations should provide a good basis for modeling human brain activity during natural language comprehension. We extract transformations from the widely-studied BERT model[33,58] and use encoding models to evaluate these transformations against several other families of linguistic features in terms of predicting brain activity during natural language comprehension (Fig. 1B, C). We find that the transformations perform comparably to the embeddings, and generally outperform both non-contextual embeddings and classical syntactic annotations—suggesting that the contextual information extracted from surrounding words is surprisingly rich. In fact, transformations at earlier layers of the model account for more unique variance in brain activity than the embeddings themselves. Finally, we disassemble these transformations into the functionally specialized computations performed by individual attention heads. We find that certain properties of the heads, such as look-back distance, dominate the mapping between headwise transformations and cortical language ears. We also find that, for some language regions, headwise transformations that preferentially encode certain linguistic dependencies also better predict brain activity.

## Results

We adopted a model-based encoding framework[59–61] in order to map Transformer features onto brain activity measured using fMRI while subjects listened to naturalistic spoken stories (Fig. 1A). Our principal theoretical interest lies in the transformations, because these are the components of the model that introduce contextual information extracted from other words into the current word. In these models, any syntactic or compositional structure linking one word to another must emerge from the transformations implemented by the attention heads. Although the transformations may approximate certain syntactic operations, they do not explicitly disentangle syntax from the meaning of words and can incorporate content-rich contextual relationships. Given that the cortical language network also does not appear to cleanly differentiate syntax and other linguistic features[46,62–66], we theorized that the transformations may provide a useful basis for modeling neural activity during natural language processing.

We pursued two core questions: First, what is the efficacy of these transformations in predicting brain activity relative to both embeddings and other types of language features? We hypothesized that (a) the transformations would predict brain activity better than other types of language features; and (b) that the transformations would map onto cortical language areas in a more layer-specific way than embeddings, given that the embeddings accumulate contextual information across layers. Second, we address the exploratory question of whether the functional specialization observed in the transformations implemented by individual attention heads maps onto brain activity in a structured way. We operationalize "shared functional specialization" as a correspondence wherein headwise transformations that preferentially encode linguistic dependencies also better predict brain activity.

We spatially downsampled the brain data according to a fine-grained functional atlas comprising 1000 cortical parcels[67], which were grouped into a variety of regions of interest (ROIs) spanning early auditory cortex to high-level language areas[68]. Parcelwise encoding models were estimated using banded ridge regression with three-fold cross-validation for each subject and each story[69]. Phonemes, phoneme rate, word rate, and a silence indicator were included as confound variables during model estimation and discarded during model evaluation[30]. Encoding models were evaluated by computing the correlation between the predicted and actual time series for test partitions; correlations were then converted to the proportion of a noise ceiling estimated via intersubject correlation (ISC)[70] (Fig. S1).

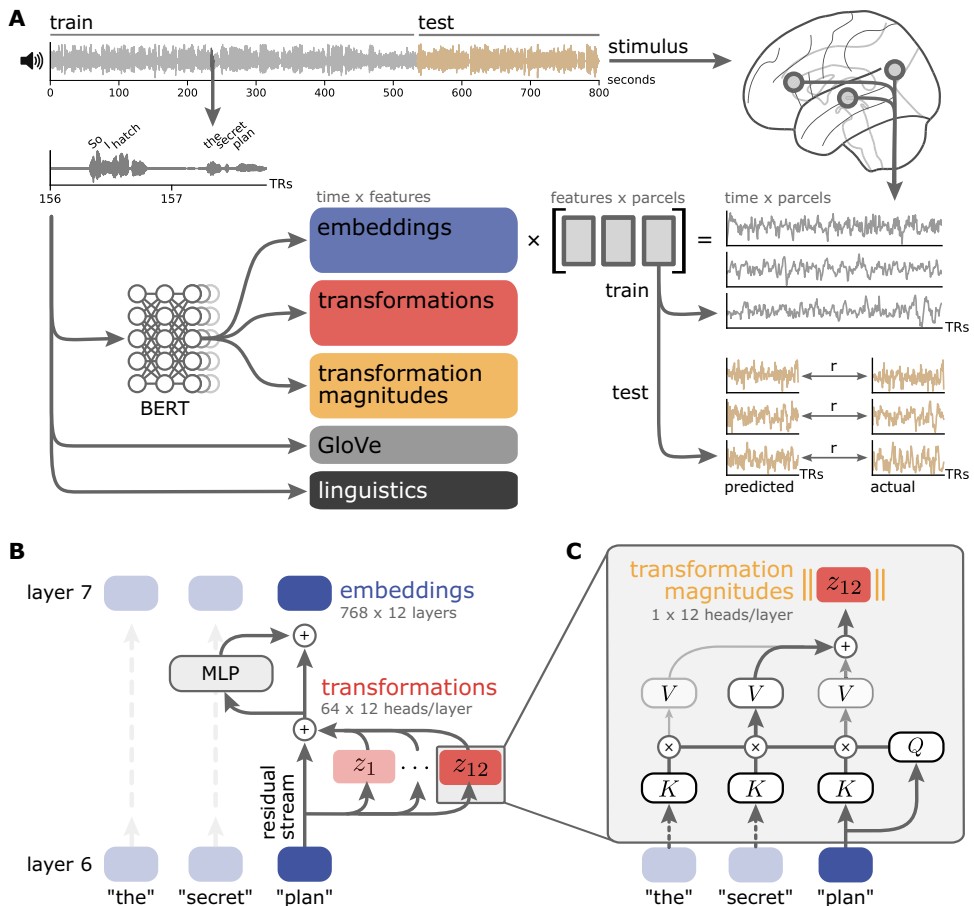

**Fig. 1 | Encoding models for predicting brain activity from the internal components of language models. A** Various features are used to predict fMRI time series acquired while subjects listened to naturalistic stories. Based on the stimulus transcript, we extracted classical linguistic features (e.g., parts of speech; black), non-contextual semantic features (e.g., GloVe vectors; gray), and internal features from a widely studied Transformer model (BERT-base). The encoding models are estimated from a training subset of each story using banded ridge regression and evaluated on a left-out test segment of each story using three-fold cross-validation. Model predictions are evaluated by computing the correlation between the predicted and actual time series for the test set. **B** We consider two core components of the Transformer architecture at each layer (BERT-base and GPT-2 each have 12 layers): embeddings (blue) and transformations (red). Embeddings represent the contextualized semantic content of the text. Transformations are the output of the self-attention mechanism for each attention head (BERT-base and GPT-2 have 12 heads per layer, each producing a 64-dimensional vector). Transformations capture the contextual information incrementally added to the embedding in that layer. Finally, we consider the transformation magnitudes (yellow; the L2 norm of each attention head's 64-dimensional transformation vector), which represent the overall activity of a given attention head. MLP: multilayer perceptron. **C** Attention heads use learned matrices to produce content-sensitive updates to each token. For a single input token ("plan") passing through a single head (layer 7, head 12), the token vector is multiplied by the head's learned weight matrices (which are invariant across inputs) to produce query ($Q$), key ($K$), and value ($V$) vectors. The inner product between the query vector for this token ("plan", $Q$) and the key vector ($K$) for each other token yields a set of "attention weights" that describe how relevant the other tokens are to "plan." These "attention weights" are used to linearly combine the value vectors ($V$) from the other tokens. The summed output is the transformation for each head (here, $Z_{12}$). The results from each attention head in this layer are concatenated and added back to the token's original representation. Figure made using Nilearn, Matplotlib, seaborn, and Inkscape.

## Functional anatomy of a Transformer model

We take BERT-base[33] as a representative Transformer model due to its success across a broad suite of NLP tasks and the burgeoning literature studying its internal representations[39–42,58,71–76]. Words are first assigned non-contextual static embeddings, which are submitted as input to the model. Unlike recurrent architectures[34,35,57], which process word embeddings serially, BERT considers long "context windows" of up to 512 tokens and processes all words in the context window in parallel. Like most Transformer architectures, BERT is composed of a repeated self-attention motif: 12 sequential layers, each consisting of 12 parallel attention heads. A single attention head consists of three separate query, key, and value matrices. At each layer, the input word embeddings are multiplied by these matrices to produce a query, key, and value vector for each word. For an individual word, self-attention then computes the dot product of that word's query vector with the key vector from all words in the context window, resulting in a vector of attention weights quantifying the relevance of other words. These

weights are used to produce a weighted sum of all value vectors, which is the output of the attention head. We refer to these 64-dimensional vectors as the "transformation" produced by that attention head. The transformation vectors are then concatenated across all heads within a layer and passed through a feed-forward module (a multilayer perceptron; MLP) to produce a fused 768-dimensional output embedding, which serves as the input for the next layer (Fig. 1B). Although the transformations at a given layer are "cued" by the embedding arriving from the previous layer, they are not derived from this embedding; similarly, the transformations are nonlinearly fused with the content of the output embedding (see "Transformer-based features" in Methods for further details). Embeddings are sometimes referred to as the "residual stream": the transformations at one layer are added to the embedding from the previous layer, so the embeddings accumulate previous computations that subsequent computations may access[77].

The self-attention mechanism can be thought of as a "soft," or weighted, key-value store. Each word in the input issues a query which

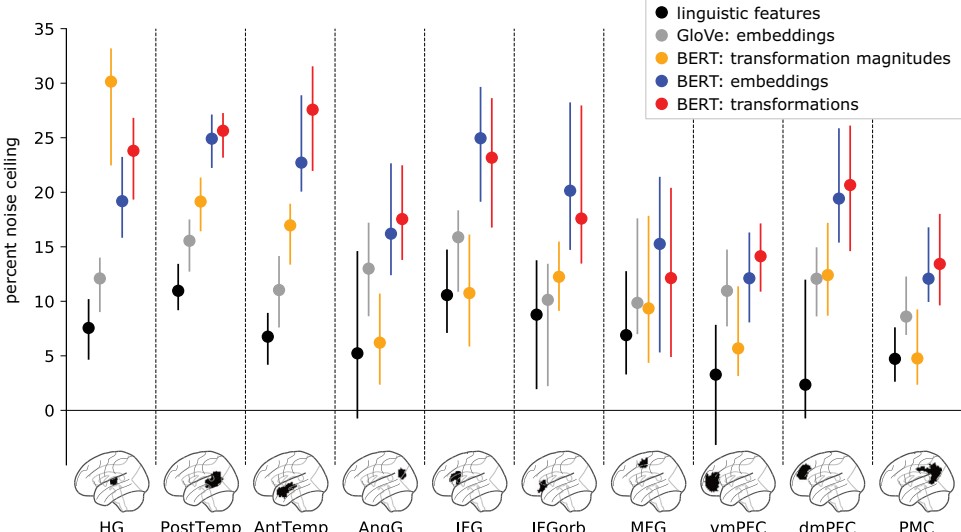

**Fig. 2 | Comparing three classes of language models across cortical language areas.** We used encoding models to evaluate the performance of three different classes of language models: classical linguistic features, non-contextual word embeddings (GloVe), and contextual Transformer features (BERT). Among the Transformer features, embeddings capture the contextual semantic content of words, transformations capture the contextual transformations that yield these embeddings, and transformation magnitudes capture the non-semantic contribution of each head to a given token. For this analysis, Transformer features were concatenated across all heads and layers. Only left-hemisphere language ROIs are included here; right-hemisphere language ROIs yielded qualitatively similar results (Fig. S2). Model performance is evaluated in terms of the percent of a noise ceiling estimated using intersubject correlation (see "Noise ceiling estimation" in "Methods" for further details; Fig. S1). Markers indicate median performance across participants ($N = 63$ subjects) and error bars indicate 95% bootstrap confidence intervals. HG Heschl's gyrus, PostTemp posterior temporal lobe, AntTemp anterior temporal lobe, AngG angular gyrus, IFG inferior frontal gyrus, IFGorb orbital inferior frontal gyrus, MFG middle frontal gyrus, vmPFC ventromedial prefrontal cortex, dmPFC dorsomedial prefrontal cortex, PMC posterior medial cortex. Source data are provided as a Source Data file. Figure made using Nilearn, Matplotlib, seaborn, and Inkscape.

is checked against the keys of all context words. However, unlike a traditional key-value store (which would only return a single, exact query-key match), the attention head first computes how well the query matches with all keys (the attention weights), and returns a weighted sum of each word's value based on how closely they match. The query, key, and value matrices are learned, and recent work has revealed an emergent functional specialization where specific attention heads approximate certain syntactic relationships[55,56]. For example, prior work has discovered that a specific head in BERT reliably represents the direct object relationship between tokens[56]. In a phrase such as "hatch the secret plan," the "plan" token would attend heavily to the "hatch" token and update its representation accordingly. More precisely, $\mathbf{q_{plan}}^T \cdot \mathbf{k_{hatch}}$ will yield a large attention weight from "plan" to "hatch." As a result, the transformation for "plan," $\mathbf{z_{plan}}$, will be heavily weighted towards $\mathbf{v_{hatch}}$ (Eq. 1). This allows the attention head to update the "plan" token's representation to reflect the fact that it is being "hatched" (as opposed to being executed, revised, or abandoned; or being used in another sense entirely, e.g., an architectural blueprint).

### Transformer-based features outperform other linguistic features

Before disassembling the transformations into specialized circuit computations, we first evaluated how well the transformations, considered in aggregate, perform against other commonly studied language features in predicting brain activity. The transformations are the conduit by which syntactic and contextual information are incorporated into the current word. We hypothesized that this rich contextual information would put the transformations on par with the embeddings in terms of predicting brain activity, and that the transformations would outperform other linguistic features. To evaluate these hypotheses, we compared the encoding performance of features from three families of language models: (1) classical linguistic features comprising parts of speech and syntactic dependencies; (2) GloVe

word embeddings[37] that capture the "static" or non-contextual meanings of words; and (3) contextualized Transformer features extracted from BERT—namely, layer-wise embeddings, transformations, and transformation magnitudes. For each TR, we appended the words from the preceding 20 TRs as context. BERT was allowed to perform bidirectional attention across the tokens in these 21 TRs, after which the context tokens were discarded, and the TR tokens were averaged to obtain the Transformer features. To summarize the overall performance of these different Transformer features, we concatenated features across all heads and layers, allowing the regularized encoding model to select the best-performing combination of those features across the entire network. We use these features to predict response time series in cortical parcels comprising ten language ROIs ranging from early auditory cortex to high-level, left-hemisphere language areas (Fig. 2; see Fig. S2 for right-hemisphere results). Based on prior work[50,78], we expected the BERT embeddings to outperform the GloVe and linguistic features. We further hypothesized that the set of transformations would perform on par with the embeddings. Finally, we hypothesized that the transformation magnitudes, intended to capture the relative contribution of each head, abstracted away from the semantic content, would more closely match the performance of classical linguistic features.

First, we confirmed that Transformer embeddings and transformations outperform classical linguistic features in most language ROIs ($p < 0.005$ in HG, PostTemp, AntTemp, AngG, IFG, IFGorb, vmPFC, dmPFC, and PMC for both embeddings and transformations; permutation test; FDR corrected; Table S1). In a follow-up analysis, we found that embeddings and transformations also outperform an "effort" metric extracted from a state-of-the-art incremental sentence parser[79] in almost all language ROIs (Fig. S3). Contextual Transformer embeddings also outperform non-contextual GloVe embeddings across several ROIs in keeping with prior work[50,78]. Interestingly, transformation magnitudes outperform GloVe embeddings and classical linguistic features in lateral temporal areas but not in higher-level language

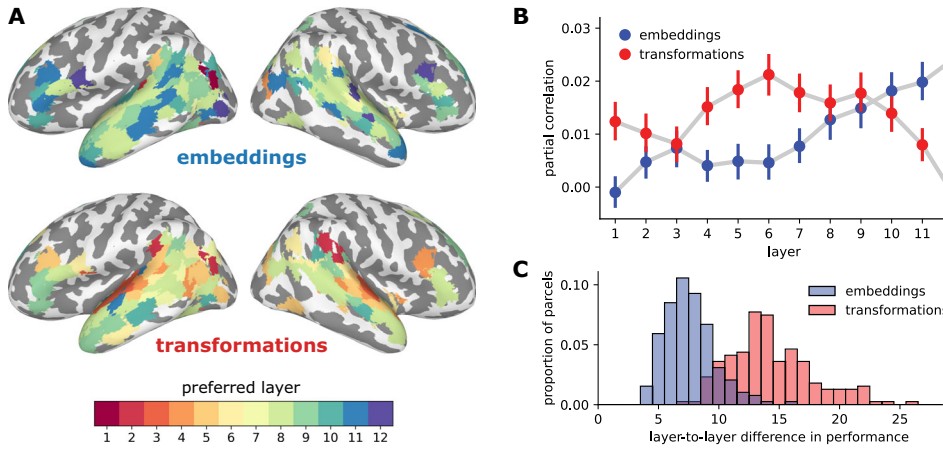

**Fig. 3 | Layer preferences for embeddings and transformations. A** Layer preferences are visualized on the cortical surface for embeddings (upper) and transformations (lower). While most cortical parcels prefer the final embedding layers, the transformations reveal a cortical hierarchy of increasing layer preference. Only cortical parcels with encoding performance greater than 20% of the noise ceiling for both embeddings and transformations are included for visualization purposes. The same color map for preferred layer is used for both embeddings and transformations. **B** Partial correlations between brain activity and model-based predictions derived from embeddings (blue) and transformations (red). For each layer, we measured the correlation between transformation-based predictions and brain activity while controlling for the embedding-based predictions (and vice versa). Partial correlations at each layer were averaged across parcels in the cortical language network. Error bars denote 95% bootstrap confidence intervals across subjects ($N = 63$). **C** Distribution of the magnitude of layer-to-layer differences in encoding performance for embeddings and transformations; this metric of layer specificity is quantified as the L2 norm of the first differences between encoding performance for neighboring layers. Transformations (red) yield more layer-specific deviations in performance than embeddings (blue). Source data are provided as a Source Data file. Figure made using SUMA, Matplotlib, seaborn, and Inkscape.

areas: for example, transformation magnitudes outperform GloVe embeddings in posterior and anterior temporal areas, but this pattern is reversed in the angular gyrus. Finally, we found that the transformations roughly match the embeddings across all ROIs. This overall pattern of results was replicated in the autoregressive GPT-2 model (Fig. S4). Note that despite yielding similar encoding performance, the embeddings and transformations are fundamentally different; for example, the average TR-by-TR correlation between embeddings and transformations across both stimuli is effectively zero ($-0.004 \pm 0.009$ SD), and the embeddings and transformations are not correlated across layers (Fig. S5). This is expected, given that embeddings and transformations reside in different feature spaces, where the transformations are translated into the embedding space by an MLP. The embeddings and transformations also yield visibly different TR-by-TR representational geometries (Fig. S6), and the transformations have considerably higher temporal autocorrelation than the embeddings (Fig. S7). As a control analysis, we evaluated these features in a non-language ROI (early visual cortex) and found that no models captured a significant amount of variance (Fig. S8). Overall, these findings suggest that the transformations capture a considerable proportion of variance of neural activity across the cortical language network and motivate more detailed treatment of their functional properties.

**Layerwise performance of embeddings and transformations**
We next segregated the Transformer features into separate layers. There is an important theoretical distinction in the layer-by-layer structure of the embeddings and transformations arising from the architecture of the network. The embeddings encode the meaning of the current word and become increasingly contextualized from layer to layer[55]. Residual connections allow the embeddings to propagate and accumulate information across layers[77]. The transformations, on the other hand, capture the "updates" to the embedding at each layer—derived from other words in the surrounding context. The transformations are largely independent from layer to layer (Fig. S9) and produce more layer-specific representational geometries (Figs. S10 and S11). Based on these distinct computational roles, we

hypothesized that the transformations would map onto the brain in a more layer-specific way than the embeddings.

First, we found that, across language ROIs, the performance of contextual embeddings increased roughly monotonically across layers, peaking in late-intermediate or final layers (Figs. S12A and S13), replicating prior work[43,47,80,81]. Interestingly, this pattern was observed across most ROIs, suggesting that the hierarchy of layerwise embeddings does not cleanly map onto a cortical hierarchy for language comprehension. Transformations, on the other hand, seem to yield more layer-specific fluctuations in performance than embeddings and tend to peak at earlier layers than embeddings (Figs. S12B, C and S14).

We next visualized layer preference across cortex—that is, which layer yielded the peak performance for a given cortical parcel (Fig. 3A). Across language parcels, the average performance (across participants) for transformations peaked at significantly earlier layers than performance for embeddings (mean preferred transformation layer = 7.2; mean preferred embedding layer = 8.9; $p < 0.001$, permutation test). To evaluate the unique contributions of transformations- and embedding-based predictions at each layer, we performed a partial correlation analysis: we measured the correlation between transformation-based predictions and brain activity while controlling for the embedding-based predictions (and vice versa; Fig. 3B). We found that transformation-based predictions capture more unique variance at earlier layers than embedding-based predictions; embeddings, on the other hand, accumulate information over time and capture the most unique variance at later layers. Finally, we quantified the magnitude of difference in predictive performance from layer to layer and found that transformations have larger differences in performance between neighboring layers (mean layerwise embedding difference = 7.6, mean layerwise transformation difference = 14.3; $p < 0.001$, permutation test; Fig. 3C). These results recapitulate the progression of layer specificity reported in the literature[43,47,80,81], and suggest that the computations implemented by the transformations are more layer-specific than the embeddings. However, we contend that these layerwise trends provide only a coarse view of functional specialization: individual attention heads perform strikingly diverse linguistic operations even within a given layer.

## Interpreting transformations via headwise analysis

Does the emergent functional specialization of internal computations in the language model reflect functional specialization observed in the cortical language network? To begin answering this question, we first directly examined how well classical linguistic features—indicator variables identifying parts of speech and syntactic dependencies—map onto cortical activity. Despite a large body of work using experimental manipulations of phrases and sentences to dissociate syntax from semantics and localize syntactic operations in the brain[82–88], results have been mixed, leading some authors to suggest that syntactic computations may be fundamentally entangled with semantic representation and distributed throughout the language network[46,63–66]. Along these lines, we found that classical linguistic features are poor predictors of brain activity and do not provide a good basis for examining functional specialization in the cortical language network in the context of naturalistic narratives (Figs. 2, S3, and S15).

BERT's training regime has been shown to yield an emergent headwise functional specialization for particular linguistic operations[55,56]. BERT is not explicitly instructed to represent syntactic dependencies, but nonetheless seems to learn coarse approximations of certain linguistic operations from the structure of real-world language[56].

We split the transformations at each layer into their functionally specialized components—the constituent transformations implemented by each attention head. In the following analyses, we leverage these functionally specialized headwise transformations to map between syntactic operations and the brain in a way that retains some level of interpretability, but respects the contextual nature of real-world language. Note that the embeddings incorporate information from all the transformations at a given layer (and prior layers), and therefore cannot be meaningfully disassembled in this way. We trained an encoding model on all transformations and then evaluated the prediction performance for each head individually, yielding an estimate of how well each head predicts each cortical parcel (or headwise brain prediction score). For each attention head, we also trained a set of decoding models to determine how much information that head contains about a given syntactic dependency (or headwise dependency prediction score; Fig. S16). In line with prior work[55,56], we empirically confirmed that the transformations at certain attention heads preferentially encode certain linguistic dependencies in our stimuli (Table S2).

For each classical syntactic dependency, we first identified the attention head that best predicts that dependency (for example, head 11 of layer 6 best predicts the direct object dependency, Fig. S15). We compared the encoding performance for each classical dependency with the encoding performance for the headwise transformation that best predicts that dependency. We found that the head most associated with a given dependency generally outperformed the dependency itself (Fig. S15; see Fig. S17 replication using functionally specialized heads derived from larger text corpora[56]). This confirmed our expectation that the dense, emergent headwise transformations are better predictors of brain activity than the sparse, classical linguistic indicator variables. Note that the headwise transformations are considerably higher-dimensional (64 dimensions) than the corresponding one-dimensional dependency indicators. However, we found that even after reducing a given transformation to a single dimension that best predicts the corresponding dependency, the one-dimensional transformation still better predicts brain activity than the dependency itself (Fig. S18). We found that these one-dimensional transformation time series are highly correlated with the corresponding dependency indicators, but reflect a continuous, graded representation of the dependency over the course of a narrative (Fig. S19). Although the computations performed by these heads approximate particular syntactic operations, they capture a more holistic relationship between words in the context of the narrative.

That is, the transformations do not simply indicate the presence, for example, of a direct object relationship; rather, they capture an approximation of the direct object relationship in the context of the ongoing narrative.

Critically, BERT does not just learn to approximate certain classical syntactic operations; it learns a rich multiplicity of linguistic and contextual relations from natural language, a subset of which can be said to approximate classical syntactic labels[58]. With this in mind, we pursued a data-driven analysis to summarize the contributions of all headwise transformations across the entire language network (Fig. 4A). We first obtained the trained encoding model for all transformations (Fig. 1, red) and averaged the regression coefficients (i.e., weight matrices) assigned to the transformation features across subjects and stimuli. To summarize the importance of each head for a given parcel, we segmented the learned weight matrix from the encoding model for that parcel into the individual attention heads at each layer and computed the L2 norm of the headwise encoding weights. This results in a single value for each of the 144 heads reflecting the magnitude of each head's contribution to encoding performance at each parcel; these vectors capture each parcel's "tuning curve" across the attention heads. In order to summarize the contribution of headwise transformations across the language network, we aggregated the headwise encoding weight vectors across parcels in language ROIs and used principal component analysis (PCA) to reduce the dimensionality of this transformation weight matrix across parcels, following Huth and colleagues[30]. This yields a basis set of orthogonal (uncorrelated), 144-dimensional weight vectors capturing the most variance in the headwise transformation weights across all language parcels; each head corresponds to a location in this low-dimensional brain space. The first two principal components (PCs) accounted for 92% of the variance in weight vectors across parcels, while the first nine PCs accounted for 95% of the variance. A given PC can be projected into (i.e., reconstructed in) the original space of cortical parcels, yielding a brain map where positive and negative values indicate positive and negative transformation weights along that PC (Fig. S20). Visualizing these PCs directly on the brain reveals that PC1 ranges from strongly positive values (red) in bilateral posterior temporal parcels and left lateral prefrontal cortex to widespread negative values (blue) in medial prefrontal cortex (Fig. 4B). PC2 ranged from positive values (red) in prefrontal cortex and left anterior temporal areas to negative values (blue) in partially right-lateralized temporal areas (Fig. 4C). Note that the polarity of these PCs is consistent across all analyses, but is otherwise arbitrary.

We next examined whether there is any meaningful structure in the "geometry" of headwise transformations in this reduced-dimension cortical space. To do this, we visualized several structural and functional properties of the heads in two-dimensional projections of the language network. We first visualized the layer of each head in this low-dimensional brain space and found a layer gradient across heads in PC1 and PC2 (Fig. 4D). PCs 9, 5, and 1 were the PCs most correlated with layer assignment with $r = 0.45$, 0.40, and 0.26, respectively (Figs. S20 and S21). Intermediate layers were generally located in the negative quadrant of both PC1 and PC2 (corresponding to blue parcels in Fig. 4B, C; e.g., posterolateral temporal cortex), with early and late layers located in the positive quadrant (red parcels). For each head, we next computed the average backward attention distance across stimuli. PCs 2, 1, and 3 were the PCs most correlated with backward attention distance with $r = 0.65$, 0.20, 0.19, respectively (Figs. S20 and S22). We observed a strong gradient of look-back distance increasing along PC2 (Fig. 4E); that is, prefrontal and left anterior temporal parcels (red parcels in Fig. 4C) correspond to heads with longer look-back distances. Note that the upper quartile of headwise attention distances exceeds 30 tokens, corresponding to look-back distances on the scale of multiple sentences. We also found that the functionally specialized heads previously reported in the literature[56]

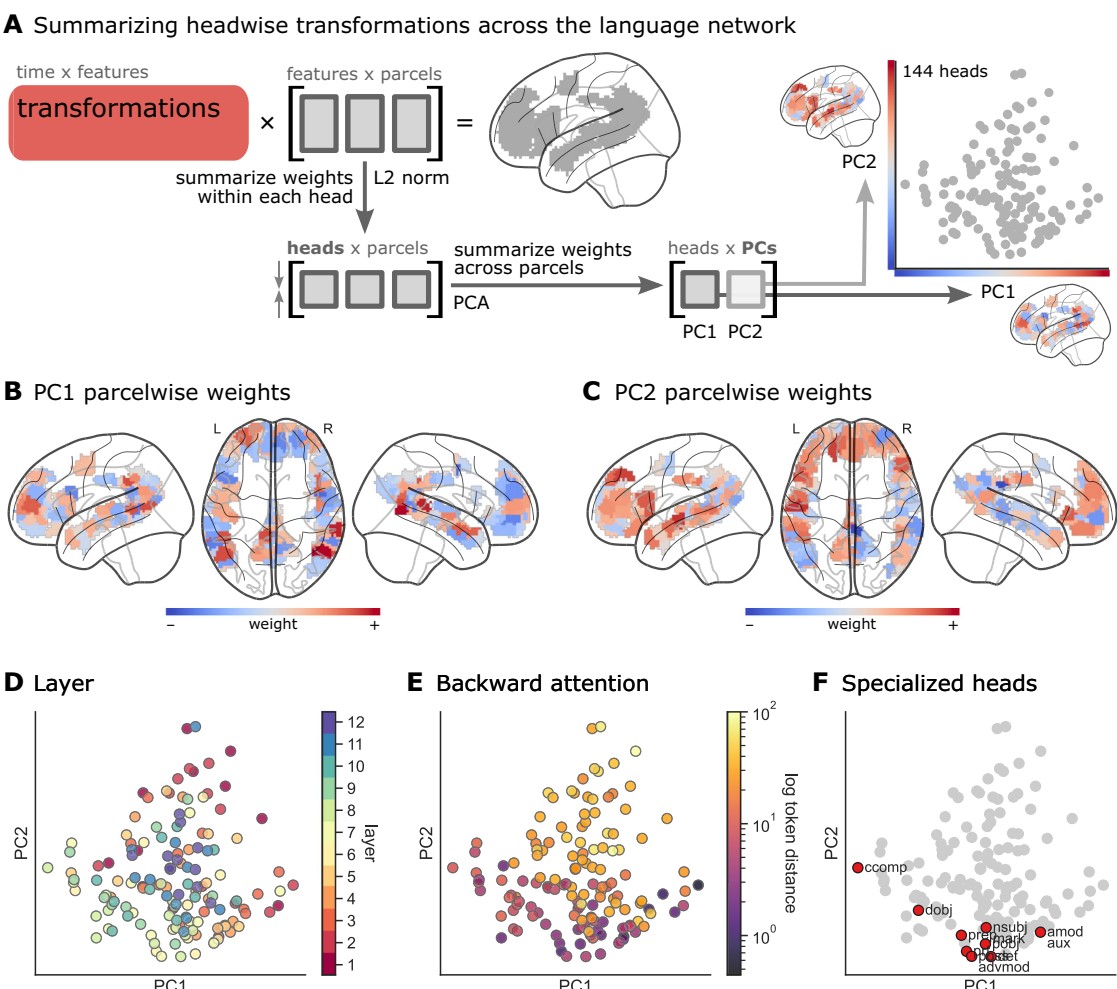

**Fig. 4 | Headwise transformations in a low-dimensional brain space. A** We applied PCA to the weight vectors for transformation encoding models across language parcels, effectively projecting the transformation weights into a low-dimensional brain space[30]. We first obtain the parcelwise weight vectors for the encoding model trained to predict brain activity from BERT transformations. This transformation weight matrix is shaped (768 features × 12 layers) 9,216 features × 192 language parcels. We use the L2 norm to summarize the weights within each head, reducing this matrix to (12 heads × 12 layers) 144 heads × 192 language parcels. We next summarize these headwise weights across language parcels using PCA. At right, we visualize the headwise transformation weights projected onto the first two PCs. Each data point corresponds to one of 144 heads. Furthermore, each PC can be projected back onto the language network (see Fig. S24 for a control analysis). **B, C** PC1 and PC2 projected back onto the language parcels; red indicates positive weights, and blue indicates negative weights along the corresponding PC. **D** Heads colored according to their layer in BERT in the reduced-dimension space of PC1 and PC2. **E** Heads colored according to their average backward attention distance in the story stimuli (look-back token distance is colored according to a log-scale). **F** Heads highlighted in red have been reported as functionally specialized by Clark and colleagues[56]. Source data are provided as a Source Data file. Figure made using Nilearn, Matplotlib, seaborn, and Inkscape.

span PC1 and cluster at the negative end of PC2 (corresponding to intermediate layers and relatively recent look-back distance; Fig. 4F). Finally, we visualized the headwise dependency prediction scores in this low-dimensional brain space and observed gradients in different directions along PC1 and PC2 for different dependencies (Fig. S23). Note that none of the aforementioned structural or functional properties of the heads that we visualize in this low-dimensional brain space are derived from the brain data; that is, the encoding models do not "know" the layer or backward attention distance of any given head.

We next devised a control analysis to test whether the structure observed in Fig. 4 depends on the organization of transformation features into functionally specialized heads. We shuffled the coefficients assigned to transformation features by the encoding model across heads within each layer of BERT. We then repeated the same analysis: we segmented the shuffled transformation features back into "pseudo-heads," computed the L2 norm of the coefficients within each pseudo-head, and applied PCA across language parcels. This perturbation disrupts the emergent functional grouping of transformation features into particular heads observed in the unperturbed model.

After this perturbation, the first two PCs accounted for only 17% of variance across language parcels (reduced from 92% in the unperturbed model). PCs were dramatically less correlated with layer assignment (maximum r across PCs reduced from 0.45 to 0.25) and look-back distance (maximum r across PCs reduced from 0.65 to 0.26). Finally, this perturbation abolished any visible geometry of layer, look-back distance, or headwise dependency decoding in the low-dimensional projection onto PCs 1 and 2 (Fig. S22). This control analysis indicates that the structure observed in Fig. 4 does not arise trivially, and results from the grouping of transformation features into functionally specialized heads; transformation features map onto brain activity in a way that systematically varies head by head, and shuffling features across heads (even within layers) disrupts this structure (Fig. S24). We observed similar trends in GPT-2 (Fig. S25); interestingly, however, look-back distance was most highly correlated with PC1 (r = 0.77), and layer was highly correlated with PC3 (r = 0.60).

Finally, to quantify the correspondence between the syntactic information contained in a given head and that head's prediction performance in the brain, we computed the correlation, across

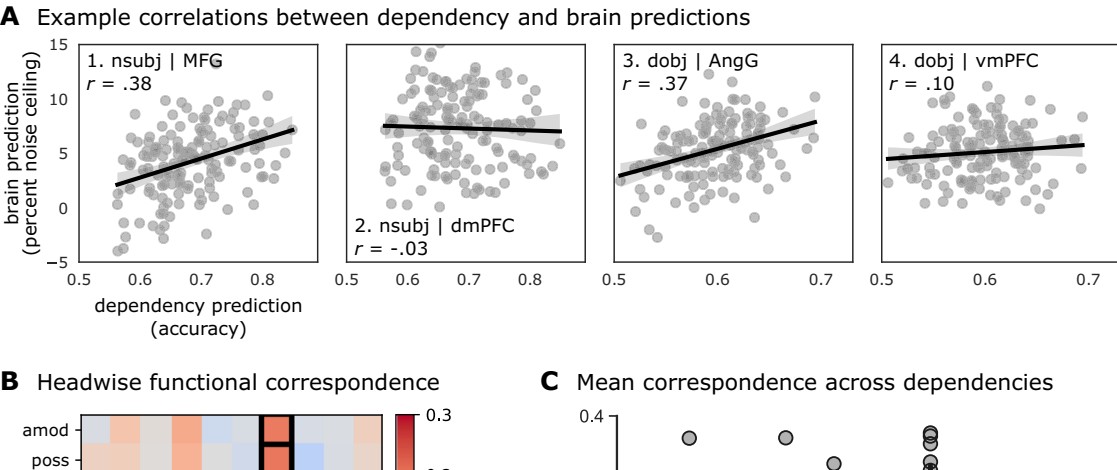

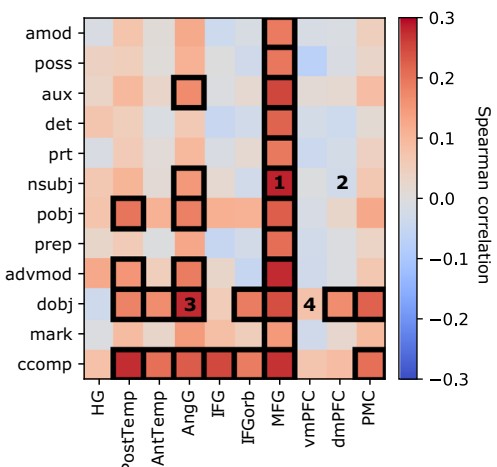

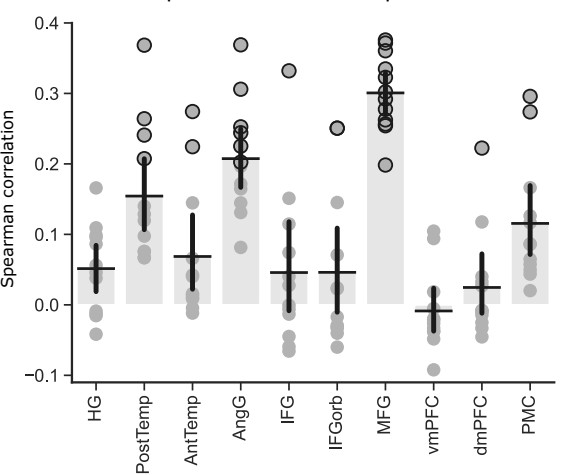

**Fig. 5 | Correspondence between headwise brain and dependency predictions.**
**A** Correlation between headwise brain prediction and dependency prediction scores, for example, ROIs and dependencies; nominal subject (nsubj) in MFG and dmPFC, direct object (dobj) in AngG and vmPFC (see Fig. S26 for correlations plotted for all ROIs and dependencies). Each point in the scatter plot represents the dependency prediction ($x$ axis) and brain prediction ($y$ axis) scores for each of the 144 heads. Brain prediction scores reflect cross-validated encoding model performance evaluated in terms of the percent of a noise ceiling estimated using inter-subject correlation. Dependency prediction scores reflect the classification accuracy of a cross-validated logistic regression model trained to predict the occurrence of a given linguistic dependency at each TR from the 64-dimensional transformation vector for a given attention head. Each of these plots corresponds to a labeled cell in the dependencies-by-ROI correlation matrix in **B**. Error bands around the line of best fit represent 95% bootstrapped confidence intervals.
**B** Correlation between headwise brain prediction and dependency prediction scores for each language ROI and syntactic dependency. Dependencies ($y$ axis) are ordered by their token distance; e.g., the adjectival modifier (amod) spans fewer tokens on average than the clausal complement (ccomp; see "Methods" for details). Cells with black borders contain significant correlations as determined by a two-tailed permutation test in which we shuffle assignments between headwise dependency prediction scores and brain prediction scores across heads (FDR controlled at $p < 0.05$). Labeled cells correspond to the example correlations in **A**. Dependencies are described in Table S5. **C** We summarize the brain–dependency prediction correspondence for each ROI by averaging across syntactic dependencies (i.e., averaging each column of **B**). Error bars indicate 95% bootstrap confidence intervals around the mean across $N = 12$ dependencies. Each data point denotes a dependency, and black borders indicate dependencies with significant correspondence. Source data are provided as a Source Data file. Figure made using Matplotlib, seaborn, and Inkscape.

attention heads, between the brain prediction and dependency prediction scores (Fig. 5A, B). We repeated this analysis for each syntactic dependency and the parcels comprising each language ROI (Fig. S26). Headwise correspondence between dependencies and ROIs indicates that attention heads containing information about a given dependency also tend to contain information about brain activity for a given ROI—thus linking that ROI to the computation of that dependency. We found that the correspondence between brain prediction and dependency prediction scores varied considerably across ROIs. For example, in posterior superior temporal cortex, we observed headwise prediction correspondence for clausal complement (ccomp), direct object (dobj), and preposition object (pobj) relations. In IFG, on the other hand, headwise prediction correspondence was observed only for the clausal complement relation. Headwise prediction correspondence was high in the angular gyrus and MFG across dependencies: for example, attention heads that predict the existence of a nominal subject relationship (nsubj) also tend to predict the MFG, but not the dmPFC; heads that predict direct object (dobj) tend to predict the

angular gyrus, but this relationship is weaker in the vmPFC. In the case of MFG, this is consistent with prior work implicating MFG in both language comprehension and more general cognitive demand (e.g., working memory[89,90]). Collapsing across dependencies highlights the discrepancy between ROIs (Fig. 5C). The angular gyrus and MFG display a relatively high correspondence; in contrast, the vmPFC and dmPFC display virtually no correspondence. While transformations explain significant variance in these ROIs at the scale of the full model (Fig. 2), individual layers (Fig. S12), and individual heads (Fig. 5A), their prediction performance for the brain does not correlate with their prediction performance for classic syntactic dependencies—suggesting that the shared information between transformations and certain ROIs may be semantic in nature or reflect contextual relationships beyond the scope of classical syntax.

To ensure the observed correspondence does not arise trivially, we designed two control analyses. In the first control analysis, we shuffled the transformation features across heads within each layer of BERT and then performed the same functional correspondence

analysis. This control analysis tests whether the observed correspondence depends on the functional organization of transformation features into particular heads. Perturbing the functional grouping of transformation features into heads reduced both brain and dependency prediction performance and effectively abolished the headwise correspondence between dependencies and language ROIs (Fig. S27). In the second control, we supplied our stimulus transcripts to an untrained, randomly initialized BERT architecture, extracted the resulting transformations, and evaluated headwise correspondence with the brain. Headwise functional correspondence was similarly abolished for the untrained model (Fig. S28). This indicates that the correspondence is not simply a byproduct of the model's architecture or our experimental stimuli, but depends in part on the model learning certain statistical structures in real-world language. Finally, to ensure that our approach generalizes across models, we replicated this analysis in GPT-2. GPT-2 yielded higher correspondence values, particularly in IFG, but with less specificity across ROIs (Fig. S29). Note that GPT-2 and BERT have different architectures and different training regimes; given that the models have similar encoding performance overall (Fig. S4), the observed differences in functional correspondence highlight the sensitivity of our headwise analytic framework.

## Discussion

In the current study, we evaluated the transformations implemented by the attention heads of BERT against several other kinds of linguistic features in terms of predicting brain activity. There are three properties of the transformations that motivated our analyses. First, the transformations are the component of the model that allow information to flow between words: all syntactic, compositional, and contextual relations across words are generated by the transformations. Unlike classical syntactic annotations and syntactic parsers, however, the transformations do not explicitly disentangle syntax from word meaning and will learn to approximate whatever contextual relations are useful for predicting words in real-world text. Second, unlike the embeddings, which accumulate information from layer to layer, the transformations encode "updates" to the embedding at each layer, derived from the surrounding context. Third, the transformations at a given layer can be disassembled into the functionally specialized computations performed by individual attention heads. These properties are of particular theoretical interest for understanding how neural systems like the brain construct context-rich meaning across individual words in natural language. While a growing body of work has used the Transformer architecture[32,33] to model the neural basis of human language processing[43,44,47,50,51,78,80], we contribute a complementary perspective on how to relate these models to human brain activity.

We found that the transformations provide a surprisingly good basis for modeling human brain activity during natural language comprehension. The transformations perform on par with the embeddings and outperform other linguistic features across most language ROIs, suggesting that the contextual information the transformations extract from surrounding words is surprisingly rich. We also found that the transformations at earlier layers of the model account for more unique variance than the embeddings, and map onto cortical language areas in a more layer-specific fashion. Examining the contribution of headwise transformations to encoding performance reveals gradients in a low-dimensional cortical space that reflect certain structural and functional properties of the headwise transformations, including layer and look-back distance. Finally, we quantified the correspondence between headwise predictions of brain activity and syntactic dependencies for a variety of cortical language areas and dependencies, and found that headwise transformations that best predict certain dependencies also best predict certain ROIs (e.g., PostTemp, AngG, MFG). We show that this correspondence does not arise arbitrarily, but depends on the functional grouping of

transformations into heads, and on the model's architecture and training regime.

To build an intuition for why the transformations may provide complementary insights to the embeddings, we can compare Transformers to convolutional neural networks (CNNs) commonly used in visual neuroscience[61,91]. CNNs typically reduce dimensionality across layers[92,93], putting pressure on the model to gradually discard task-irrelevant, low-level information and retain only high-level semantic content. In contrast, popular Transformer architectures maintain the same dimensionality across layers. Thus Transformer embeddings can aggregate information (from context words) across layers, such that later layers tend to contain the most information[55] (albeit over-specialized for a particular downstream training objective; i.e., the cloze task for BERT). In this light, it is unsurprising that encoding performance tends to peak at later embedding layers. Indeed, unlike the structural correspondence between CNN layers and the visual processing hierarchy[61,94,95], Transformer embeddings are highly predictive but relatively uninformative for localizing stages of language processing. Unlike the embeddings, the transformations reflect updates to word meanings at each layer. Encoding models based on the transformations must "choose" a step in the contextualization process, rather than "have it all" by simply using later layers.

Despite the formal distinction between syntax and lexico-semantics in linguistics, the neural computations supporting human language may not so cleanly dissociate syntactic and semantic processing[46,62–66], especially during natural language comprehension, where syntax and semantics are typically intertwined. In practice, we found that classical linguistic features (i.e., parts of speech, syntactic dependencies, parser effort) are relatively poor predictors of brain activity during natural language comprehension (Figs. 2, S3 and S18). Although Transformer models implicitly learn syntactic and compositional operations in order to produce well-formed linguistic outputs, these emergent structures are generally entangled with semantic content[41,42,96]. Indeed, much of our theoretical interest in the transformations stems from the observation that, although they approximate syntactic operations to some extent, they can also more expressively code for content- and context-rich relationships across words. We attribute the relatively strong prediction performance of the transformations to this rich contextual information.

While the transformations used in the current analysis capture syntactic and contextual operations entangled with semantic content, the transformation magnitudes can serve to disentangle syntax and semantics. Prior work has sought to isolate the two using artificial stimuli and vector subtraction on the embeddings[46]; the transformation magnitudes instead reduce the transformations down to the "activation" of individual attention heads. Insights from NLP[56] suggest this metric, which circumvents the stimulus representation entirely, nonetheless contains an emergent form of syntactic information. Our comparison to classical linguistic features (Figs. 2 and S3) suggests this is the case: transformation magnitudes outperform classical linguistic features in temporal language areas, and perform comparably elsewhere. Interestingly, the transformation magnitudes also outperform the non-contextual word embeddings in temporal areas (Fig. 2), while this relationship is reversed in angular gyrus, a putative high-level convergence zone for semantic representation[97].

We adopted a data-driven approach to describe the properties of the transformations predominant in brain activity during naturalistic language processing. The gradients depicted in Fig. 4 are most easily understood as summarizing the language network's top two (orthogonal) tuning curves across the transformations implemented by 144 attention heads in BERT. The distance attention heads tend to "look back" in the narrative (Fig. 4, PC2), and to a lesser extent layer assignment (Fig. 4, PC1), accounted for considerable variance in the mapping between the headwise transformation and the cortical language network. We found that the functional properties of the

headwise transformations do, in fact, map onto certain cortical localization trends previously reported in the literature. For example, Fig. 4 revealed that posterior temporal areas assign higher weights to heads at earlier layers (positive values along PC1) with shorter look-back distance (negative values along PC2), consistent with previous work suggesting that posterior temporal areas perform early-stage syntactic (and lexico-semantic) processing[15,98–100]. Headwise correspondence in posterior temporal cortex was high for the ccomp and dobj dependencies (Fig. 5), which are involved in resolving the meaning of verb phrases, corroborating prior work implicating posterior temporal areas in verb–argument integration[86,101,102]. We also found that left-lateralized anterior temporal and anterior prefrontal cortices were associated with longer look-back attention distances (positive values along PC2; Fig. 4), suggesting that these regions may have longer temporal receptive windows[14,103–105] and compute longer-range contextual dependencies, including event- or narrative-level relations[106–111]. Interestingly, the IFG (pars opercularis and triangularis; i.e., Broca's area) was not strongly associated with heads specialized for particular syntactic operations (Fig. 5B, C), despite being well-predicted by both BERT embeddings and transformations (Fig. 2). There are several possible explanations for this: (1) the natural language stimuli used here may not contain sufficient syntactic complexity to tax IFG; (2) the cortical parcellation used here may yield imprecise functional localization of IFG[112,113]; and (3) the IFG may be more involved in language production than passive comprehension[114].

Although the transformations yield relatively strong prediction performance, they are not positioned to serve as mechanistic models of cortical language processing—just because a model predicts brain activity better than other models does not make it a good model[115–117]. That said, the transformations occupy a relatively underexplored space among theories of language processing: the transformations capture content-rich contextual relationships across words that inflect the meaning of the current word. This is in contrast to lexical-semantic models that capture only the meaning of individual words and parsing models that capture syntactic relationships among words without contextual meaning. While the embeddings (and resulting word predictions) are the "final product" of the contextualization process in Transformer-based models, the transformations capture the contextual processes that build up to this end point. More broadly, the human language system and end-to-end large language models do—to some extent—share common computational principles aimed at producing well-formed, context-sensitive linguistic outputs[50,118]: these models represent the unique meaning of words in context; these models are not "given" linguistic structures, but rather learn linguistic structure from natural language using a biologically-plausible, self-supervised objective function; these structures are encoded in high-dimensional embedding spaces across relatively simple computing elements; finally, these models better reproduce human-like performance than prior generations of models across a variety of natural language tasks[31,33,119–122].

Our results suggest several future lines of research. Prior work has explored different Transformer architectures[78,80] aiming to establish a structural mapping between Transformers and the brain. Toward this end, training "bottlenecked" Transformer models that successively reduce the dimensionality of linguistic representations—similar to CNNs—may produce more hierarchical embeddings and provide a better structural mapping onto cortical language circuits[123]. Second, the current work sidesteps the acoustic and prosodic features of natural speech[124,125]; the models we used operate on sequences of tokens in text and do not encode finer-grained temporal features of speech. Future work, however, may benefit from models that extract high-level contextual semantic content directly from the temporally-resolved speech signal (in the same way that CNNs operate directly on pixel values[126–129]). Third, we found that, although BERT and GPT-2 perform similarly when both mapping embeddings and transformations onto

brain activity[78,80], they differ in terms of headwise correspondence. This suggests that headwise analysis may be sensitive to differences in model–brain correspondence that are obscured when considering only the embeddings. Finally, current neurobiological models of language highlight the importance of long-range fiber tracts connecting the nodes of the language network[130–132]; we suspect that future language models with more biologically-inspired circuit connectivity may provide insights not only into functional specialization but also functional integration across specialized modules.

Transformer-based large language models like BERT and GPT obtain state-of-the-art performance on multiple NLP tasks. BERT's attention heads are functionally specialized and learn to approximate classical syntactic operations in order to produce contextualized natural language[55,56]. The rapidly developing field of BERTology[58] seeks to characterize this emergent functional specialization. In both language models and the human language network, emergent functional specialization likely reflects both architectural constraints and the statistical structure of natural language[133–136]. In this work, we took a first step toward bridging between BERTology's insights and language processing in the brain. Although we do not find a direct one-to-one mapping between attention heads, linguistic dependencies, and cortical areas, our findings suggest that certain trends in functional organization—such as a gradient of increasing contextual look-back distance—may be shared. Mapping the internal structure of large language models to cortical language circuits can bring us closer to a mechanistic understanding of human language processing, and may ultimately provide insights into how and why this kind of functional specialization emerges in both large language models and the brain.

## Methods

### Experimental data

Models were evaluated on two story datasets from the publicly available "Narratives" collection of fMRI datasets acquired while subjects listened to naturalistic spoken stories[137]. Code used to analyze the data is available at the accompanying GitHub repository: https://github.com/tsumers/bert-brains. The "Slumlord" and "Reach for the Stars One Small Step at a Time" dataset includes 18 subjects (ages: 18–27 years, mean age: 21 years, 9 reported female) and comprises two separate stories roughly 13 min (550 TRs) and 2600 words in total. The "I Knew You Were Black" dataset includes 45 subjects (ages: 18–53 years, mean age: 23.3 years, 33 reported female); the story is roughly 13 min (534 TRs) long and contains roughly 1500 words. All participants provided informed, written consent prior to data collection in accordance with experimental procedures approved by Princeton University Institutional Review Board. All functional MRI datasets were acquired with a 1.5 s TR[137], and were organized in compliance with the Brain Imaging Data Structure[138]. The number of TRs in our analyses was determined by the duration of the naturalistic spoken story stimuli; e.g., the "I Knew You Were Black" story told live by Carol Daniel for the Moth Radio Hour (https://themoth.org/stories/i-knew-you-were-black) was 800 s (13 min, 20 s) long, corresponding to 534 TRs with a 1.5-second TR.

Preprocessed MRI data were obtained from the Narratives derivatives release[137]. Briefly, the following preprocessing steps were applied to the functional MRI data using fMRIPrep 20.0.5[139]: susceptibility distortion correction (using fMRIPrep's fieldmap-less approach), slice-timing correction, volume registration, and spatial normalization to MNI space (MNI152NLin2009cAsym template). Confound regression was then implemented using 3dTproject[140] in AFNI 19.3.0 with the following confound regressors: six head motion parameters, the first five aCompCor components from CSF and from white matter[141], cosine bases for high-pass filtering (cutoff: 128 s), and first- and second-order polynomial trends. Non-smoothed functional data were used for all analyses in the current study. To harmonize datasets with differing spatial resolution and reduce computational demands, we resampled

all functional data to a fine-grained 1000-parcel cortical parcellation derived from intrinsic functional connectivity[67]. That is, time series were averaged across voxels within each parcel to yield a single average response time series per parcel (within each subject and story dataset).

We constructed a set of 10 ROIs intended to span the cortical hierarchy for language and narrative processing from low-level sensory areas (e.g., HG) to high-level association areas (e.g., PMC). First, we extracted language ROIs from Fedorenko and colleagues[68]: PostTemp, AntTemp, AngG, IFG, IFGorb, and MFG. We roughly defined each ROI as the set of parcels from the 1000-parcel atlas in which over 50% of voxels in the parcel overlapped with voxels assigned to the ROI in MNI space. At the bottom of the hierarchy, we added the early auditory cortex (Heschl's gyrus; HG), extracted from the Harvard-Oxford Atlas. At the upper end of the hierarchy, we added vmPFC, dmPFC, and PMC ROIs, manually defined from the Schaefer parcellation, to capture the representation of higher-level narratives features and events[108,142]. These ROIs encompass most of the brain areas with reliable, stimulus-driven activity when subjects listen to spoken stories[137].

All figures were created by the authors using free and open-source software: Nilearn[143], SUMA[140], Matplotlib[144], seaborn[145], and Inkscape.

### Baseline language features

Language model representations were derived from the time-locked phoneme- and word-level transcripts available in the Narratives dataset[137]. Words were assigned to the fMRI volumes based on the ending timestamps; e.g., if a word began in one TR and ended in the following TR, it was assigned to the second TR. The following low-level acoustic and linguistic features were extracted for each TR to serve as confound variables in subsequent analyses: (a) the number of words per TR, (b) number of phonemes per TR, and (c) a binary vector indicating the presence of individual phonemes per TR. These are the same confound variables used in ref. 30.

We extracted part-of-speech and dependency relations to serve as classical linguistic features. These features were annotated using the en_core_web_lg (v2.3.1) model from spaCy (v2.3.7[146]). For each TR, we created a binary feature vector across all parts-of-speech/dependency relations, indicating whether or not a given part-of-speech/dependency relation appeared within that TR. Part of speech (e.g., noun, verb, adjective) describes the function of each word in a sentence. We used 14 part-of-speech labels: pronoun, verb, noun, determiner, auxiliary, adposition, adverb, coordinating conjunction, adjective, particle, proper noun, subordinating conjunction, numeral, and interjection (Table S4). A dependency relation describes the syntactic relation between two words. For each word, a parser defines another word in the same sentence, called the "head," to which the word is syntactically related; the dependency relation describes the way the word is related to its head. We used 25 dependency relations: nsubj, ROOT, advmod, prep, det, pobj, aux, dobj, cc, ccomp, amod, compound, acomp, poss, xcomp, conj, relcl, attr, mark, npadvmod, advcl, neg, prt, nummod, and intj (Table S5; https://github.com/clir/clearnlp-guidelines/blob/master/md/specifications/dependency_labels.md). For visualization, we focus on 12 dependencies with particularly high correspondence to particular heads reported by Clark and colleagues[56].

We also used a modern combinatory categorial grammar (CCG) to capture symbolic syntactic operations in sentence processing[79]. CCGs are incremental parsers that explicitly model syntactic structure, but with more human-like expressiveness than widely-used context-free grammars (CFGs). We used code and examples provided by Stanojević and colleagues (https://github.com/stanojevic/ccgtools) to parse the transcript of the "I Knew You Were Black" stimulus and extract their metric of parsing effort. We extracted the scalar parsing effort metric for right-branching, left-branching, and left-branching-with-revealing operations, resulting in three effort scores for each word in the transcript. We summed each score across words within a given TR to downsample the word-level effort scores to the temporal resolution of the fMRI acquisition.

Finally, to serve as a baseline for Transformer-based language models, we used GloVe vectors[37], which capture the "static" semantic content of a word across contexts. Conceptually, GloVe vectors are similar to the vector representations of text input to BERT prior to any contextualization applied by the Transformer architecture. We obtained GloVe vectors for each word using the en_core_web_lg model from spaCy, and averaged vectors for multiple words occurring within a TR to obtain a single vector per TR.

### Transformer self-attention mechanism

While language models such as GloVe[37] assign a single "global" or "static" embedding (i.e., meaning) to a given word across all contexts, the Transformer architecture[32] introduced the self-attention mechanism, which yields context-specific representations. Just as convolutional neural nets[93,147] use convolutional filters to encode spatial inductive biases, Transformers use self-attention blocks as a sophisticated computational motif or "circuit" that is repeated both within and across layers. Self-attention represents a significant architectural shift from the sequential processing of language via recurrent connections[34] to simultaneously processing multiple tokens. Variations on the Transformer architecture with different dimensionality and training objectives currently dominate major tasks in NLP, with BERT[33] and GPT[31] being two of the most prominent examples.

There is an ongoing debate as to whether autoregressive transformers (e.g., GPT[31,148]) or bidirectional transformers (e.g., BERT[33]) are more appropriate models for predicting brain activity[116]. In this work, we chose BERT as a more plausible model for narrative comprehension. This is because BERT's "bidirectional" attention allows later words in a sentence to affect the meaning of earlier words, whereas GPT's "causal" attention does not. To understand the implications, consider the example shown in Fig. 1 above, containing the words "the secret plan." GPT is autoregressive, using a "causal" (rather than "bidirectional") attention mechanism. This means that information can only flow forwards in time: the representation of "plan" can be updated based on "secret", but the representation of "secret" cannot be retroactively updated based on "plan." In contrast, BERT allows bidirectional attention within the context window, so the two words can affect each others' representations. Humans clearly can and do operate bidirectionally: we are able to update our representation of earlier words based on later ones (e.g., cataphora[149]). Given our focus on the contextualization process itself (i.e., transformation vectors), we chose BERT as the more realistic model; however, we replicate our key findings in GPT-2 to showcase the generalizability of our approach.

Self-attention operates as follows. A single attention head consists of three separate (learned) matrices: a query matrix, a key matrix, and a value matrix, each of dimensionality $d_{model} \times d_{head}$, where $d_{model}$ indicates the dimensionality of the model's embedding layers, and $d_{head}$ indicates the dimensionality of the attention head. Input word vectors are multiplied by each of these three matrices independently, producing a query, key, and value vector for each word. To determine the contextualized representation for a given word vector, the self-attention operation takes the dot product of that word's query vector with the key vector from all words. The resulting values are then scaled and softmaxed, producing the "attention weights" for that word.

Formally, for a Transformer head of dimensionality $d_{head}$ and sets of query, key, and value vectors forming matrices **Q, K, V**, the self-attention mechanism operates as follows:

$$Attention(\mathbf{Q},\mathbf{K},\mathbf{V}) = softmax(\frac{\mathbf{Q} \times \mathbf{K}}{\sqrt{d_{head}}})\mathbf{V} \qquad (1)$$

The $i$th token "attends" to tokens based on the inner product of its query vector $\mathbf{Q}_i$ with the key vectors for all tokens, $\mathbf{K}$. When the query vector matches a given key, the inner product will be large; the softmax ensures the resulting "attention weights" sum to one. These attention weights are then used to generate a weighted sum of the value vectors, $\mathbf{V}$, which is the final output of the self-attention operation (Eq. 1). We refer to the attention head's output as the "transformation" produced by that head. Each attention head produces a separate transformation for each input token.

State-of-the-art Transformer architectures scale up the core self-attention mechanism described above in two ways. First, multiple attention heads are assembled in parallel within a given layer ("multi-headed attention"). For example, BERT-base-uncased[33], used in most of our analyses, contains 12 attention heads in each layer. The 12 attention heads are each 64-dimensional and act in parallel to produce independent 64-dimensional "transformation" vectors. These vectors are concatenated to produce a single 768-dimensional vector which is then passed to the feed-forward module. Second, BERT base consists of 12 identical layers stacked on top of each other, allowing the model to learn a hierarchy of transformations for increasingly complex contextualization[55]. The full model has 12 layers × 12 heads = 144 total attention heads and a total of 110 million learned parameters. The original input vectors are analogous to the static GloVe vectors; however, as they pass through successive layers in the model, repeated applications of the self-attention mechanism allow their meanings to contextualize each other.

The phenomenal success of Transformer-based models has generated an entire sub-field of NLP research, dubbed "BERTology," dedicated to reverse-engineering their internal representations[42,58]. Researchers have linked different aspects of the model to classic NLP concepts, including functional specialization of individual "attention heads" to syntactic dependencies[56], representational subspaces corresponding to parse trees[74], and an overall layerwise progression of representations that parallels traditional linguistic pipelines[55]. Our approach builds on this work, using internal Transformer features as a bridge from these classical linguistic concepts to brain data.

## Transformer-based features

We used three separate components from the Transformer models to predict brain activity. The first features we extract are the layerwise "embeddings," which are the de facto Transformer feature used for most applications, including prior work in neuroscience[78]. The embeddings represent the contextualized semantic content, with information accumulating across successive layers as the Transformer blocks extract increasingly nuanced relationships between tokens[55]. As a result, embeddings have been characterized as a "residual stream" that the attention blocks at each layer "write" to and "read" from. Later layers often represent a superset of information available in earlier layers, while the final layers are optimized to the specific pretraining task. Prior work using such models typically finds that the mid-to-late layers best predict brain activity[78,80].

The second set of features we extract are the headwise "transformations" (Eq. 1), which capture the contextual information introduced by a particular head into the residual stream prior to the feedforward layer (MLP). These features are the unique component of the transformer responsible for propagating information between different tokens (see ref. 77). Consequently, the transformations play a privileged role in constructing meaning from the intricate relationships between words: any contextual information that passes into a single word's representation is incorporated by way of transformations computed at some head[150]. In other words, it is simultaneously the earliest dense representation to result from the attention operation at a given layer (unlike the sparse attention matrices themselves) and the final head-wise representation (unlike MLP activations, which operate on the full residual stream after each head's transformations have been concatenated back together). While the transformations represent the most crucial component of the self-attention mechanism, to our knowledge, they have not been previously studied in human neuroscience.

Intuitively, it may seem as if the transformations are to some extent redundant with the embedding at the previous layer, or the resulting embedding passed to the subsequent layer. The transformations in layer $x$ are not computed from the embedding at layer $x-1$ in a straightforward way. Rather, the transformations at layer $x$ are the result of the interplay between the key-query-value ($k$-$q$-$v$) vectors, which are themselves a function of the embedding at layer $x-1$. The learned weights at each attention head specify a projection from the embedding at layer $x-1$ to a set of $k$-$q$-$v$ components, which in turn determine a nonlinear function for pulling in and combining contextual information from other tokens. Thus, although the resulting transformations at layer $x$ share the same dimensionality with the embedding at $x-1$, they encode fundamentally different kinds of information.

The embedding is cumulative—it carries both the original semantic content from layer 0 (the initial token embedding), as well as a linear combination of contextual information incorporated by transformations at prior layers. The transformations can instead be thought of as encoding context-appropriate "adjustments" or "diffs". These adjustments are added linearly into the embedding passed along from the previous layer, effectively sculpting the embedding to respect the context. In fact, the embedding at layer $x-1$ can pass through the attention heads largely unchanged via the so-called "residual stream"; the model learns when and how each transformation should adjust the embedding-based on context[77].

These adjustments are added to the embedding, effectively sculpting the embedding to respect the context. The transformations are not natively "aligned" with the embedding; they are passed through another nonlinear transformation—the MLP—that translates the transformations into the embedding space in order to add them to the embedding at layer $x$. This step effectively fuses the contextual information derived from other words with the content of the current word embedding. Thus, the adjustments implemented by the transformations are ostensibly "contained" in the new embedding at layer $x$, but they are nonlinearly fused with the content of the previous layer.

Finally, the third set of features we extract is the "transformation magnitudes," which are the L2 norm of each attention head's transformation. This is effectively a "univariate" metric for how active each attention head was: how much its update "moved" the word representation, without any information about the direction of its influence. Because the transformation magnitudes lack direction, they cannot encode any semantic information: it is not possible to discern how a word's meaning was changed, only how much it was changed. Prior work has shown that individual attention heads learn to implement particular syntactic relationships[56]; therefore, the transformation magnitudes provide information about the relationships between tokens in the TR divorced from the semantic content of the TR. These transformation magnitudes may correlate with low-level features if particular word roles tend to be distributed at specific positions in natural speech. We found that the time series of transformation magnitudes are strongly correlated ($r \approx 0.8$) with word rate and phoneme rate in our spoken story stimuli. Note, however, that when fitting the encoding models for transformation magnitudes (and other feature sets), we included word rate and phoneme rate (as well as phoneme indicators) as confound variables in a separate band (to ensure they receive their own optimal hyperparameter). Interestingly, we also found that the transformation magnitudes are related to part of speech: we used least squares regression to predict the time series of transformation magnitudes at each TR from the 14-dimensional part-of-speech indicator time series, resulting in a moderate out-of-sample correlation of $r \approx 0.4$.

Transformer models were accessed via the HuggingFace library[151]. We used the BERT-base-uncased model. We generated "embeddings" and "transformations" as follows. For each TR, we concatenated words assigned to that TR (3.75 words on average) together with words from the preceding 20 TRs. We chose 20 TRs as it corresponds to the preceding 30 s of the auditory story stimulus, averaging around 100 Transformer tokens. The majority of BERT's training occurred on sequences of 128 tokens, so this ensured that BERT was exposed to sequence lengths that were similar to its training distribution. We passed the resulting set of words through the Transformer tokenizer and then model. This procedure allowed information from the preceding time window to "contextualize" the meaning of the words occurring in the present TR. At each layer, we extracted the "transformations" (output of the self-attention submodule, Eq. 1) and the "embeddings" (output of the final feedforward layer) for only the words in that TR. We excluded the automatically appended "SEP" token. For each TR, this left us with a tensor of dimension $n_{layers} \times d_{model} \times n_{tokens}$, where $n_{layers}$ indicates the number of layers in the Transformer model, $d_{model}$ indicates the dimensionality of the model's embedding layers, and $n_{tokens}$ the number of tokens spoken in that TR. For BERT, this resulted in a 12 layer × 768 dimension × $n_{tokens}$ tensor. We omit the original static BERT embeddings (which are sometimes termed "Layer 0") and compare BERT layers 1–12 to the 12 transformation layers. To reduce this to a consistent dimensionality, we averaged over the tokens occurring within each TR, resulting in a 12 × 768 × 1 tensor for each TR. TRs with no words were assigned zeros. Finally, to generate "transformation magnitudes" for each TR, we averaged the "transformation" vectors over all tokens in the TR, then computed the L2 norm of each attention head's transformation vector.

To generate the "backward attention" metric (Fig. 4), we followed a procedure similar to the "attention distance" measure[152]. Unlike the previous analyses, this required a fixed number of Transformer tokens per TR. Rather than using the preceding 20 TRs, we first encoded the entire story using the Transformer tokenizer, and for each TR selected the 128 tokens preceding the end of the TR. This corresponded to ~30 seconds of the auditory story stimulus. We processed each TR and extracted each head's matrix of token-to-token attention weights (Eq. 1). We selected the token-to-token attention weights corresponding to information flow from earlier words in the stimulus into tokens in the present TR (excluding the special [SEP] token). We multiplied each token-to-token attention weight by the distance between the two tokens, and divided by the number of tokens in the TR to obtain the per-head attention distance in that TR. Finally, we averaged this metric over all TRs in the stimulus to obtain the headwise attention distances. Note that by focusing on backward attention distances for the transformations implemented by individual attention heads, we may underestimate attention distances that effectively accumulate over layers[71].

### Decoding dependency relations

In addition to using the linguistic features to predict brain activity, we used the "transformation" representations to predict dependencies. For each TR, the "transformation" consists of a $n_{layers} \times d_{model}$ tensor; for BERT, this yields a 12 layers × 768 dimension tensor. Each of the layers is thus a 768-dimensional vector, which itself consists of 12 concatenated 64-dimensional vectors, each corresponding to the output of a single attention head. These 64-dimensional headwise vectors were used as predictors. Note that this decomposition is only valid for transformations, which are the direct results of the multi-head attention mechanism; due to the feedforward layer after the multi-head attention mechanism, there is no trivial way to disassemble embeddings into headwise representations.

We used spaCy to annotate each word with a dependency label indicating whether the word is a child for the given dependency in a parse tree. For the dependency to be labeled as present, only the child in the dependency relationship has to be in the given TR. If any word occurring within a TR was labeled as a child in the dependency relationship, that TR was labeled as having the dependency present. We trained a separate decoder for each dependency to predict whether a dependency is present or not for each TR. If multiple dependencies co-occur within a TR, the separate classifiers for each dependency will each receive "present" labels for the corresponding dependency.

We performed logistic regression with the L2 penalty (implemented using scikit-learn[153]) to predict the occurrences of each binary dependency relation over the course of each story from the headwise transformations. The regularization hyperparameter was determined for each head and each dependency relation using nested five-fold cross-validation over a log-scale grid with 11 values ranging from $10^{-30}$ to $10^{30}$. Since some dependency relations are relatively rare, labels are imbalanced. We corrected for this imbalance by weighting samples according to the inverse frequency of occurrence during training and by using balanced accuracy for evaluation[154].

### Encoding model estimation and evaluation

Encoding models were estimated using banded ridge regression with three-fold cross-validation[69]. Ridge regression is a formulation of regularized linear regression using a penalty term to ensure that the learned coefficients minimize the squared L2 norm of the model parameters. This effectively imposes a multivariate normal prior with zero mean and spherical covariance on the model parameters[155]. Relative to ordinary least squares, ridge regression tends to better accommodate collinear parameters and better generalize to unseen data (by reducing overfitting). In banded ridge regression, the full model is composed of several concatenated submodels, including both features of interest (e.g., BERT features) and confound variables. The following confound variables were included based on prior work[30]: phoneme and word rate per TR, a 32-dimensional phoneme indicator matrix, and an indicator vector indicating silent TRs. Each of these submodels was assigned to a separate "band" in the banded ridge regression and thus received a different regularization parameter when fitting the full model. Specifically, we have separate bands for: the main model (e.g., BERT features), the silence indicator vector, word count and phoneme count vectors, and the phoneme indicator matrix. Columns of the predictor matrix were z-scored across TRs based on the mean and standard deviation of the training set. When evaluating the model, we generate predictions using only the main model band, discarding any confound features. For each band, we duplicated and horizontally stacked the feature space four times, comprising lags of 1, 2, 3, and 4 TRs (1.5, 3.0, 4.5, 6.0 s) in order to account for parcelwise variation in hemodynamic lag[30]. The regularization parameters were selected via a random search with 100 iterations. In each iteration, we sample regularization parameters for each band uniformly from the simplex by sampling from a dirichlet distribution (as implemented in the himalaya package[69]). We performed nested three-fold cross-validation within each training set of the outer cross-validation partition. Encoding models were fit for each of the 1,000 parcels within each subject and each story dataset. The cross-validation procedure was implemented so as to partition the time series into temporally contiguous segments. When conducting the headwise encoding analyses (Fig. 5), we also discard the learned coefficients corresponding to all heads except for the particular head of interest for prediction and evaluation on the test set[156].

Encoding model performance was evaluated by computing the Pearson correlation between the predicted and actual time series for the test partition. Correlation was used as the evaluation metric in both the nested cross-validation loop for regularization hyperparameter optimization, and in the outer cross-validation loop. For each partition of the outer cross-validation loop, the regularization parameter with the highest correlation from the nested cross-validation loop within

the training set was selected. This procedure yields a correlation value for each test set of the outer cross-validation loop for each parcel and subject. These correlation values were then averaged across cross-validation folds, and Fisher-z transformed prior to statistical assessment.

## Noise ceiling estimation

The raw correlation values described above depend on the signal-to-noise ratio (SNR), duration, and other particular features of the data. In order to provide a more interpretable metric of model performance, we compute the proportion of the correlation value relative to a noise ceiling—effectively, the proportion of explained variance relative to the total variance available. Acquiring replicates of brain responses to the same stimulus in the same subjects is effectively impossible under naturalistic contexts due to repetition effects; i.e., listening to a narrative for a second time does not engage the same cognitive processes as listening to the narrative for the first time[157]. To circumvent this challenge, we used ISC as an estimate of the noise ceiling[70,158]. In this approach, time series from all subjects are averaged to derive a surrogate model intended to represent the upper limit of potential model performance. For each subject, the test time series for each outer cross-validation fold is first averaged with the test time series of all other subjects in the dataset, then the test time series for that subject is correlated with the average time series. Including each subject in the average biases the noise ceiling upward, thus yielding more conservative estimates of the proportion. Note that under circumstances where individual subjects are expected to vary considerably in their functional architecture, ISC may provide a suboptimal noise ceiling. However, in the current context, we do not hypothesize or model any such differences.

## Statistical assessment

In all cases, parcelwise encoding models were estimated and evaluated within individual subjects. However, for statistical evaluation, we aggregate the model performance scores (correlation between predicted and actual test time series) derived from the subject-specific models across subjects. This approach is in contrast to "low N" studies (e.g.[30], $N = 7$), which similarly fit models within each subject, but statistically evaluate each subject separately. In our approach, fine-grained differences in cortical topographies for language across individual brains may be obscured by aggregating parcelwise results across subjects[68,112]. We compromised on these fronts for three reasons. First, in contrast to dense-sampling, low $N$ datasets (e.g.[159]), the Narratives dataset comprises larger samples of subjects (e.g., $N = 45$ for the "I Knew You Were Black" story) exposed to fewer spoken story stimuli. Second, we estimated parcelwise encoding models based on the 1000-parcel atlas by Schaefer and colleagues[67] to reduce computational demands across numerous models tested; we suspect that this spatial downsampling, while reducing the specificity of the encoding models to some degree, may also provide coarse-grained alignment when aggregating within-subject model performance scores across subjects. Third, we do not have repeated exposures to the same test set (e.g.[30]), and therefore estimate a between-subjects noise ceiling based on ISC[70]. We acknowledge that finer-grained encoding may be possible by aggregating across subjects in a more sophisticated way (e.g., using hyperalignment[20,21,160]).

To assess the statistical significance of encoding model performance, we used two nonparametric randomization tests. First, when testing whether model performance was significantly greater than zero, we used a one-sample bootstrap hypothesis test[161]. For each iteration of the bootstrap, we randomly sampled subject-level correlation values (averaged across cross-validation folds), then computed the Fisher-transformed mean across the bootstrap sample of subjects to construct a bootstrap distribution around the mean

model performance value across subjects for each parcel. We then subtracted the observed mean performance value from this bootstrap distribution, thus shifting the mean roughly to zero (the null hypothesis). Finally, we computed a one-sided $p$ value by determining how many samples from the shifted bootstrap distribution exceeded the observed mean.

Second, when comparing performance between two models, we used a permutation test. For each iteration of the permutation test, we took the subject-wise differences in performance between the two models, randomly flipped their signs, then recomputed the mean difference in correlation across subjects to populate a null distribution. We then computed a two-sided $p$ value by determining how many samples from either tail of the distribution exceeded the mean observed difference.

Statistical tests for population inference were performed by concatenating subjects across story datasets prior to randomization in order to produce one $p$ value across stories; however, randomization was stratified within each story. When assessing $p$ values across parcels or ROIs, we corrected for multiple tests by controlling the false discovery rate (FDR) at $q < 0.05$[162].

## Summarizing headwise transformation weights

To summarize the contribution of headwise transformations across the language network, we first obtained the regression coefficients—i.e., weight matrices—from the encoding model trained to predict brain activity from the BERT transformations concatenated across layers (Fig. 1, red; corresponding to the performance in Fig. 2, red). To account for the fact that the learned weights may be on different scales at different parcels (due to different regularization parameters), we first z scored the weight vectors for each parcel prior to subsequent analysis. We then averaged the parcelwise weight vectors across both subjects and stimuli. We next computed the L2 norm of the regression coefficients within each head at each layer, summarizing the contribution of the transformation at each head for each parcel. Following Huth and colleagues[30,163], we then used PCA to summarize these headwise transformation weights across all parcels in the language ROIs. This yields a reduced-dimension brain space where each data point corresponds to the transformation implemented by each of the 144 attention heads. To visualize the structure of these headwise transformations in the reduced-dimension brain space, we colored the data points according to the structural and functional properties of the heads, including their layer, backward attention distance, and dependency prediction scores.

## Reporting summary

Further information on research design is available in the Nature Portfolio Reporting Summary linked to this article.

## Data availability

The MRI data used in this study are openly available as part of the "Narratives" dataset[137], publicly available via the OpenNeuro repository at https://doi.org/10.18112/openneuro.ds002345.v1.1.4, and via Data-Lad at https://datasets.datalad.org/?dir=/labs/hasson/narratives. The Schaefer atlas was obtained from the associated GitHub repository: https://github.com/ThomasYeoLab/CBIG/tree/master/stable_projects/brain_parcellation/Schaefer2018_LocalGlobal. The language ROIs were obtained from Fedorenko and colleagues: https://evlab.mit.edu/funcloc/. The Harvard-Oxford atlas was obtained from FSL: https://fsl.fmrib.ox.ac.uk/fsl/fslwiki/Atlases. Source data are provided with this paper.

## Code availability

Code used to analyze the data is publicly available at the accompanying GitHub repository: https://github.com/tsumers/bert-brains (https://doi.org/10.5281/zenodo.10863840).

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

## Acknowledgements

S.K. was supported by NIH T32MH065214 and a Google PhD Fellowship for the duration of this work. T.S. is supported by an NDSEG Fellowship. R.H. is supported by a C.V. Starr Fellowship. S.A.N. is supported by NIH R01MH112566. This publication was supported by the Princeton University Library Open Access Fund.

## Author contributions

Conceptualization: S.K., T.R.S., R.D.H., S.A.N., T.Y. Data curation: S.A.N., S.K., T.R.S. Formal analysis: S.K., T.R.S., S.A.N., T.Y., R.D.H. Funding acquisition: T.L.G., U.H,. K.A.N.. Investigation: S.A.N., T.R.S., S.K., T.Y. Methodology: S.K., T.R.S., S.A.N., R.D.H., T.Y., A.G.. Project administration: S.A.N., R.D.H. Software: T.R.S., S.K., T.Y., S.A.N. Supervision: S.A.N., R.D.H., T.L.G., K.A.N., U.H. Visualization: S.K., S.A.N., R.D.H, T.R.S., T.Y. Writing—original draft: S.A.N., T.R.S., S.K., R.D.H. Writing—review & editing: S.A.N., T.R.S., S.K., R.D.H., K.A.N., T.L.G., U.H., A.G.

## Competing interests

The authors declare no competing interests.
