## [Peer Review File · Nature Communications]

Shared functional specialization in transformer-based language models and the human brainReviewer #1 (Remarks to the Author):

In this manuscript, Kumar et al study the similarity between representations and computations in transformer-based language models and the brain fMRI response. In particular, they propose a new approach to investigate headwise attentional computations, which they term "transformations". Their results demonstrate that transformations capture syntactic representations in complement to the more commonly used hidden layer embeddings in the transformer models. These representations predict neural response to speech-language differentially in specific cortical regions.

Overall I think this is a good paper, providing a novel perspective on the important topic of AI and the brain. The authors performed sophisticated analyses to showcase the characteristics of the transformation encoding in the brain. I do have several comments and suggestions I hope the authors could address, which I think would improve the clarity and accessibility of the results.

1. While the authors spent a considerable amount of effort trying to demonstrate the transformation encoding, it is still not entirely clear what are the distinctions between the transformations and the embeddings. To my understanding, the transformations in layer x are directly computed from the embeddings in layer $(x-1)$ through self-attention, and the transformations in layer x are then transitioned into the embeddings of layer x through an MLP. Then should we expect the transformations to be contained by the embeddings in layer (x) and layer $(x-1)$? I wonder if the transformations and the embeddings are correlated to each other or not. Furthermore, are transformations and embeddings explaining the same or different aspects of brain activity? Are their brain predictions correlated or not?

2. The authors evaluated dependency prediction using headwise transformations. The methodological details need to be described better. It is not clear how the dependencies were decoded. Is it a binary classifier or a multi-class classifier? Is it true that if a pair of words with a specific dependency co-occurred in the same TR, then the TR was labeled as positive for that dependency? What if multiple pairs of dependencies fell into the same TR?

3. It appears that the results are very sensitive to the choice of LLM. GPT-2 and BERT show vastly different patterns, e.g. Fig 5BC vs Fig S26BC. Given these discrepancies, what would be the take-home message, aside from that the headwise functional correspondence did correlate to the brain prediction in some areas in an LLM-dependent way? The authors may consider discussing this a bit more.

4. In Figure 4, the regression weights of the transformations in the neural encoding models were analyzed. Did the authors look at the distributions of the magnitude and variance in the transformations? Are the transformations normalized so that their weights can be directly compared?

5. Also regarding the cortical map shown in Figure 4. How is the PC weight mapping computed from the transformations compared to the PCA of the neural activity themselves? The latter should reflect the functional neural network structure. I am curious if the transformation-based network is associated with the underlying functional network or not. Now it seems that as a data-driven map, it lacks a clear connection to what we already know about the language network.

6. The authors showed encoding results on 10 ROIs. How are these 10 ROIs picked? Are these all the ROIs that demonstrated significant speech-response?

7. The computation of backward attention distance seems to be constrained within each layer. However the actual attention is cumulated over the layers, the current method may underestimate the actual attention distance, particularly for the later layers. (See Abnar & Zuidema 2020, Quantifying attention flow in transformers)

8. The authors performed shuffling of the coefficients as a control analysis on Page 12. I wonder if the results would be consistent if using a randomized Transformer model as a control.

Reviewer #2 (Remarks to the Author):

Summary: While a majority of studies bridging large language models and neuroscience utilize the hidden states of the LMs as predictors of brain activity, this study proposes a novel framework to instead use the intermediate transformations (namely, the attention vectors in transformers). Their approach has the advantage that many studies in NLP have shown attention vectors to be highly specific and interpretable, especially for syntactic operations. The authors show that (1) these features predict as well as the hidden states (2) (due to the network architecture) exhibit more variability across layers than the hidden states (3) predict interesting gradients across the cortical surface with respect to layer selectivity and timescale.

Overall, this paper was clear, well motivated, interesting and carefully written so as to not over-claim. I believe that it brings a novel approach to bridge AI models and neuroscience. I only have minor questions and clarifications listed below.

Questions/Comments:

- Why do the authors use group-averaging when they could have built encoding models for individual participants? Is it to facilitate the computation of ISC as a noise ceiling estimate?
- It is interesting that the transformer magnitudes outperform GloVe embeddings in low-level areas. How correlated is this feature with word rate or part-of-speech tags? I wonder if its capturing periodicity across tokens (for example, the content words in English are distributed at specific positions) that matches areas like HG but is devoid of semantic content that would be relevant for AngG.
- Since the attention vectors are passed onto an MLP, the low correlation with the hidden state is to be expected. Similarly, the residual connection would cause higher autocorrelation in the hidden state. I think the authors should note this clearly in the text:
- "Note that despite yielding similar encoding performance, the embeddings and transformations are fundamentally different; for example, the average TR-by-TR correlation between embeddings and transformations across both stimuli is effectively zero ($-.004 \pm .009$ SD), the embeddings and transformations yield visibly different TR-by-TR representational geometries (Fig. S4), and the transformations have considerably higher temporal autocorrelation than the embeddings (Fig. S5)."
- Re arguing that they are "fundamentally different": Obviously, the two representations are related to each other. But the MLP & residual connections imply that they don't encode information the same way. Moreover, BERTology has shown that while it is difficult to understand information encoded in the hidden states, individual attention heads are often interpretable + Your results show very interesting differences in encoding performance and layer hierarchy!
- I agree with the arguments presented here to support the claim that "suggest that the computations implemented by the transformations are more layer-specific than the embeddings", namely (1) the know function of different attention heads in the literature and (2) the residual connections across hidden states which would make them more similar (it would be nice to note this explicitly). However, isn't a stronger form of evidence here showing the amount of unique vs. shared variance explained by hidden states/attention vectors across different layers?
- Minor: For the analysis in Fig. 4, were the same regularization coefficient used for the different encoding models as that could give rise to arbitrary differences in scale across parcels and participants.
- I appreciate that the authors didn't overstate the results in Fig. 4-5. However, I am curious to hear how these results can be placed in the context of previous work like Huth et al., 2016 and Lerner et al., 2011 that also make predictions about cortical gradients like in Fig. 4. Since the relationship between function and ROI was based on merely correlating the dependency prediction with the encoding model weights, couldn't it be possible that these regions don't have a purely syntactic role but their function is related to this property? (I know that the authors never claimed this, just curious to know what they think) Specifically, can we expect any differences in functional predictions made by this approach based on whether an ROI is correlated with many syntactic markers like AngG/MFG vs. ROIs that don't correlate with any like vmPFC? How does this fit with more recent studies that show no dissociation between syntax and semantics?

Reviewer #3 (Remarks to the Author):

The manuscript is well-written and the figures are pleasant to follow – the topic is potentially a new take on what aspects of transformer architectures to correlate with neuroimaging data. I wanted to be excited about this work but I ended being baffled by its motivation and meaning, and later frustrated by the weak analytical argumentation (lack of comparisons/controls supporting the conclusions). The manuscript was also peppered with unsubstantiated claims (specifically what transformers learn, and if they learn linguistic structure – that is has by no means been rigorously tested and convincingly shown, yet is discussed as if it is given and not a contentious point of ongoing debate). Most mystifyingly, the manuscript ignores the large body of work by the Brennan and Hale labs that relate parser states to brain states. That was very puzzling and seemingly a huge oversight – that work *is* the literature on relating NLP models to neural data that transformer states should be compared to. See for example: Stanojević, M., Brennan, J. R., Dunagan, D., Steedman, M., & Hale, J. T. (2023). Modeling Structure-Building in the Brain With CCG Parsing and Large Language Models. *Cognitive Science*, 47(7), e13312.

At first I was very interested in this new approach of “transformations” but was ultimately disappointed in the lack of theoretical grounding for the idea of transformations. Without theory, the work is very well-executed engineering exercise – namely, the big worry here, what is the theory and the hypothesis being tested here? If the proposal is that transformers and brains are doing something “the same” or “alike,” then what, specifically, is that thing? Is the brain doing what a transformer is doing, if so, how? There are so many problems here (e.g., the brain has time, transformers don’t, copious behavioral and formal capacities, etc., nevermind training data and training protocol differences). This quote from the Abstract, might be a good way to highlight the problem: “LLM and the cortical language network may converge on similar trends of functional specialization for processing natural language”- ok, but what are those “similar trends”? How could they arise given the stunning and fundamental differences between the two systems? What would “similar trends” mean? Then, why are some parts of the transformer architecture (here, transformations and the brain regions) better fits than other parts (embeddings)? If you’re finding that some system with regularities has regularities that are more similar to another system with regularities, you’re really just saying there are correlations between regularities (which is trivially true), unless you’re interpreting those regularities to be meaningful (e.g., the two systems are doing the same thing, behaving the same way, or whatever) – for that you need a theory and a hypothesis about what is similar between the systems, why it is so, and what it means.

Relatedly, if the conclusion is, transformer states predict brain states ergo transformers and brains are doing something similar, there are two problems. First, on a high level that is trivially true – they are both processing sequential language stimuli – although with very different behaviors, outcomes, physical and formal expressive constraints. But logically, the argument falls apart; we see this in the history of science with epicycles and geocentrism – to cut a long story short, basically, geocentric models could keep adding parameters to better predict the movement of celestial bodies than heliocentric models could. Geocentric models predicted better, but the sun is at the center of our solar system. Second, given that not all aspects of the transformer predict neural data, and not all areas of the brain are predicted, why is the transformer’s failure to predict not interpreted as “doing something fundamentally different”? These two logical pitfalls have recently been discussed at length here:

Guest, O., & Martin, A. E. (2023). On logical inference over brains, behaviour, and artificial neural networks. *Computational Brain & Behavior*, 1-15.

It is hard to evaluate whether transformations are an improvement, whether that improvement has meaning or not, when they are not compared to performance by parsers (e.g., a la Brennan and Hale work, RNN-G). Another option would be to illustrate through comparison with equally sparse predictors as the “classical linguistic annotations”. For the record, POS is not a feature that in and of itself is typically used to predict neural states, so it seems unfair to use that as a benchmark. If the goal of the work is to show that transformation are better predictors than embeddings, that seems more of an engineering exercise. It could be strengthened in to a theory by theorizing as suggested above, and then by careful model evaluation across classes of existing

successful models. Now as it stands the meaning of this work is frustratingly unclear.

Minor comments

1. What is the "neurobiology of language in silico"? Is this the claim that you can study the brain by using transformers? If so, what is the argument for why you can? Correlation?
2. Time is totally different in the brain and in transformers' transformation (spoiler: there isn't any) – how were the number of TRs chosen? It seems arbitrary at the moment.
3. In Figure S17 500 TRs are plotted. Here you can clearly see that the sparsity differences between dependencies and transformations. This should be carefully explored and controlled for given the signal processing consequences. Second, how and why a transformer head best picks out a dependency type when its timeseries is so different from that of the dependency begs for a theory of how an attention head is (if it is) representing information about the dependency, and what that information is.

Reviewer 1

In this manuscript, Kumar et al study the similarity between representations and computations in transformer-based language models and the brain fMRI response. In particular, they propose a new approach to investigate headwise attentional computations, which they term “transformations”. Their results demonstrate that transformations capture syntactic representations in complement to the more commonly used hidden layer embeddings in the transformer models. These representations predict neural response to speech-language differentially in specific cortical regions.

Overall I think this is a good paper, providing a novel perspective on the important topic of AI and the brain. The authors performed sophisticated analyses to showcase the characteristics of the transformation encoding in the brain. I do have several comments and suggestions I hope the authors could address, which I think would improve the clarity and accessibility of the results.

We appreciate the reviewer’s positive assessment of the manuscript. As you will see below, these comments were very helpful in pushing us to clarify our thinking and writing.

1. While the authors spent a considerable amount of effort trying to demonstrate the transformation encoding, it is still not entirely clear what are the distinctions between the transformations and the embeddings. To my understanding, the transformations in layer x are directly computed from the embeddings in layer $(x-1)$ through self-attention, and the transformations in layer x are then transitioned into the embeddings of layer x through an MLP. Then should we expect the transformations to be contained by the embeddings in layer (x) and layer $(x-1)$? I wonder if the transformations and the embeddings are correlated to each other or not. Furthermore, are transformations and embeddings explaining the same or different aspects of brain activity? Are their brain predictions correlated or not?

Thank you for highlighting the confusion in our writing on this point. We have clarified the relationship between embeddings and transformations in several concrete ways. First, we note that transformations in layer x are not computed from the embedding at layer $x - 1$ in a direct or straightforward way. Rather, the transformations at layer x are the result of the interplay between the key-query-value ($k-q-v$) vectors, which are *themselves* a function of the embedding at layer $x - 1$. The learned weights at each attention head specify a projection from the embedding at layer $x - 1$ to a set of $k-q-v$ components, which in turn determine a nonlinear function for pulling in and combining contextual information from *other* tokens. Thus, although the resulting transformations at layer x share the same dimensionality with the embedding at $x - 1$, they encode fundamentally different kinds of information.

The embedding is *cumulative* – it carries both the original semantic content from layer 0 (the initial token embedding), as well as a linear combination of contextual information incorporated by transformations at prior layers. The transformations can instead be thought of as encoding context-appropriate “adjustments” or “diffs”. These adjustments are *added* linearly into the embedding passed along from the previous layer, effectively sculpting the embedding to respect the

context. In fact, the embedding at layer $x - 1$ can pass through the attention heads largely unchanged via the so-called “residual stream”; the model *learns* when and how each transformation should adjust the embedding based on context (Elhage et al., 2021, provide a very intuitive formulation of this process). As the reviewer notes, the transformations are not natively “aligned” with the embedding; they are passed through a nonlinear transformation—the MLP—that effectively translates the transformations into the embedding space in order to add them to the embedding at layer x . This step effectively fuses the contextual information derived from other words with the content of the current word embedding.

This is all to say that we cannot, in a straightforward way, “expect the transformations to be contained by the embeddings in layer x and layer $x - 1$ ”: the transformations themselves are “cued” by the contextual combination of all embeddings in the input context at layer $x - 1$. Thus, the adjustments implemented by the transformations are ostensibly “contained” in the embedding at layer x , but they are nonlinearly fused with the content of the embedding.

We have added a clarifying sentence to the “Functional anatomy of a Transformer” section at the beginning of Results:

Although the transformations at a given layer are “cued” by the embedding arriving from the previous layer, they are not derived from this embedding; similarly, the transformations are nonlinearly fused with the content of the output embedding (see “Transformer-based features” in Methods and Materials for further details).

And we have added new explanatory paragraphs in the “Transformer-based features” section of the Methods section:

Intuitively, it may seem as if the transformations are to some extent redundant with the embedding at the previous layer, or the resulting embedding passed to the subsequent layer. The transformations in layer x are not computed from the embedding at layer $x - 1$ in a straightforward way. Rather, the transformations at layer x are the result of the interplay between the key-query-value ($k-q-v$) vectors, which are themselves a function of the embedding at layer $x - 1$. The learned weights at each attention head specify a projection from the embedding at layer $x - 1$ to a set of $k-q-v$ components, which in turn determine a nonlinear function for pulling in and combining contextual information from other tokens. Thus, although the resulting transformations at layer x share the same dimensionality with the embedding at $x - 1$, they encode fundamentally different kinds of information.

The embedding is *cumulative* – it carries both the original semantic content from layer 0 (the initial token embedding), as well as a linear combination of contextual information incorporated by transformations at prior layers. The transformations can instead be thought of as encoding context-appropriate “adjustments” or “diffs”. These adjustments are added linearly into the embedding passed along from the previous layer, effectively sculpting the embedding to respect the context. In fact, the embedding at layer $x - 1$ can pass through the attention heads largely

unchanged via the so-called “residual stream”; the model learns when and how each transformation should adjust the embedding based on context (Elhage et al., 2021).

These adjustments are added to the embedding, effectively sculpting the embedding to respect the context. The transformations are not natively “aligned” with the embedding; they are passed through another nonlinear transformation—the MLP—that translates the transformations into the embedding space in order to add them to the embedding at layer x . This step effectively fuses the contextual information derived from other words with the content of the current word embedding. Thus, the adjustments implemented by the transformations are ostensibly “contained” in the new embedding at layer x , but they are nonlinearly fused with the content of the previous layer.

Given the complexity of these relationships, we’ve also made an effort to empirically address the reviewer’s specific questions and help readers build intuition. In a new Fig. S3, we show that the transformations are neither correlated with the embedding at layer $x - 1$ (before the transformation) nor layer x (after the transformation). Despite the matched dimensionality, the embeddings and transformations do not share a redundant feature space. We now point to this new figure early in the results section:

Note that despite yielding similar encoding performance, the embeddings and transformations are fundamentally distinct representations; for example, the average TR-by-TR correlation between embeddings and transformations for both stimuli is effectively zero ($-.004 \pm .009$ SD), and the embeddings and transformations are not correlated across layers (Fig. S5). This is expected, given that embeddings and transformations reside in different feature spaces, and the transformations are incorporated into the next layer’s embedding space through a nonlinear mapping. The embeddings and transformations also yield visibly different TR-by-TR representational geometries (Fig. S6), and the transformations have considerably higher temporal autocorrelation than the embeddings (Fig. S7).

Figure S5. Pairwise correlations between transformations and embeddings across layers. For each TR, we computed the correlation between the transformations and embedding at each pair of layers, then averaged the resulting correlation matrices. As expected, the transformations and embeddings are not strongly correlated, even at adjacent layers. Although the transformations and embeddings have matching dimensionality, they do not share the same feature space.

To address the reviewer's questions of whether the brain predictions from the transformations and embeddings are explaining different aspects of brain activity or are correlated, we ran a partial correlation analysis between the layerwise predictions of the transformations and embeddings compared to the actual brain activity (for each test set). We appreciate the reviewer suggesting this analysis because it provides a compelling view of our core results. We've added this layerwise partial correlation analysis to Fig. 3 in the main text:

Figure 3. (B) Partial correlations between brain activity and model-based predictions derived from embeddings (blue) and transformations (red). For each layer, we measured the correlation between transformation-based predictions and brain activity while controlling for the embedding-based predictions (and vice versa). Partial correlations at each layer were averaged across parcels in the cortical language network. Error bars denote 95% bootstrap confidence intervals across subjects

We also added the following passage the Results section:

To evaluate the unique contributions of transformations- and embedding-based predictions at each layer, we performed a partial correlation analysis: we measured the correlation between transformation-based predictions and brain activity while controlling for the embedding-based predictions (and vice versa; Fig. 3B). We found that Transformation-based predictions capture more unique variance at earlier layers than embedding-based predictions; embeddings, on the other hand, accumulate information over time and capture the most unique variance at later layers.

2. The authors evaluated dependency prediction using headwise transformations. The methodological details need to be described better. It is not clear how the dependencies were decoded. Is it a binary classifier or a multi-class classifier? Is it true that if a pair of words with a specific dependency co-occurred in the same TR, then the TR was labeled as positive for that dependency? What if multiple pairs of dependencies fell into the same TR?

Thanks for pointing this out—we are happy to clarify. In the dependency decoding analysis, we used a binary classifier: spaCy annotates each word with a unique dependency label—i.e. whether the word is a child for the given dependency in a parse tree—and a separate decoder is trained for each dependency to predict whether that dependency is present or not for each TR (see the spaCy documentation). For the dependency to be labeled as present, only the child in the dependency relationship has to be in the given TR. If *any* word occurring within a TR was labeled as a child in the dependency relationship, that TR was labeled as having the dependency present. Note that we are not classifying *which* dependencies are present across TRs, but running separate classifiers for each dependency. If multiple dependencies co-occur within a TR, the separate classifiers for each dependency will each receive “present” labels for the corresponding dependency. We have modified the Methods section of the revised manuscript to clarify these details:

We used spaCy to annotate each word with a dependency label indicating whether the word is a child for the given dependency in a parse tree. For the dependency to be labeled as present, only the child in the dependency relationship has to be in the given TR. If any word occurring within a TR was labeled as a child in the dependency relationship, that TR was labeled as having the dependency present. We trained a separate decoder for each dependency to predict whether a dependency is present or not for each TR. If multiple dependencies co-occur within a TR, the separate classifiers for each dependency will each receive “present” labels for the corresponding dependency.

3. It appears that the results are very sensitive to the choice of LLM. GPT-2 and BERT show vastly different patterns, e.g. Fig 5BC vs Fig S26BC. Given these discrepancies, what would be the take-home message, aside from that the headwise functional correspondence did correlate to the brain prediction in some areas in an LLM-dependent way? The authors may consider discussing this a bit more.

The reviewer correctly notes that certain results (e.g. Figs. S25 and S29) differ for different LLMs (BERT versus GPT-2). We view this as a potential strength of our approach. There are surprisingly few published papers directly comparing how well models like BERT and GPT-2 predict brain activity. A couple examples: First, Schrimpf et al., 2021, compare overall performance in predicting brain activity for BERT, GPT-2, and a variety of other models (Fig. 2). They find that both models perform well, but GPT-2 outperforms BERT in the *Pereira2018* fMRI dataset comprising short text passages presented one sentence at a time. Note, however, two caveats: (a) larger variants of BERT (e.g. *bert-large-uncased*) performed comparably to GPT-2 on *Pereira2018*; and (b) BERT and GPT-2 performed comparably on the *Blank2014* dataset comprising longer-form naturalistic spoken stories.

Second, Caucheteux and King (2022) compared masked (e.g. BERT) and causal (e.g. GPT-2) language models and found nearly identical results using MEG (Fig. 4, panels a and e, in Caucheteux & King, 2022), with the nominal exception that brain-prediction performance for masked models peaked at earlier somewhat earlier layers than causal models.

Our goal in discussing these examples is to highlight that prior work focusing on layerwise embeddings has not convincingly shown strong differences between these models. Even in the current manuscript, we see that when evaluating both embeddings and transformations, BERT and GPT-2 perform similarly across most of the language network (Fig. S4). Our hope in examining headwise functional specialization (Figs. 4 and 5) was to develop more sensitive ways for exploring possible correspondences between model and brain. The relevant take-home message we believe is the following: Examining how model features map onto brain activity using only embeddings and at the coarse granularity of layers will tend to obscure more fine-grained model–brain relationships; we hope our approach will encourage people to consider correspondence at the granularity of the internal computations implemented at each head *within* a given layer.

We have added a passage to the Discussion in the revised manuscript in hopes of clarifying this point:

Third, we found that, although BERT and GPT-2 perform similarly when both mapping embeddings and transformations onto brain activity (Schrimpf et al., 2021; Caucheteux & King, 2022), they differ in terms of headwise correspondence. This suggests that headwise analysis may be sensitive to differences in model–brain correspondence that are obscured when considering only the embeddings.

4. In Figure 4, the regression weights of the transformations in the neural encoding models were analyzed. Did the authors look at the distributions of the magnitude and variance in the transformations? Are the transformations normalized so that their weights can be directly compared?

When fitting the encoding models, each column of the transformations (and all other features) were z-scored across TRs. The mean and standard deviation used for z-scoring were calculated based on the training set and then also applied to the test set. This ensures that the transformations were on similar scales in the regression analysis and allows us to compare the resulting weights. Reviewer 2 also commented that the weight vectors at each parcel may be on different scales due to different regularization coefficients. To account for this, in the original manuscript we also z-scored the learned weights prior to the analyses for Fig. 4.

We've added a sentence to the "Encoding model estimation" section of the Methods to clarify the first part of this comment:

Columns of the predictor matrix were z-scored across TRs based on the mean and standard deviation of the training set.

In line with Reviewer 2’s comment, we also added a clarifying sentence to the “Summarizing headwise transformation weights” in the revised Methods text:

To account for the fact that the learned weights may be on different scales at different parcels (due to different regularization parameters), we first z-scored the weight vectors for each parcel prior to subsequent analysis.

To answer the reviewer’s initial question, here we plot the layer-by-layer mean and variance in the transformations across both tokens and heads within a layer. The magnitude and variance of the transformations appear to be similar across layers.

5. Also regarding the cortical map shown in Figure 4. How is the PC weight mapping computed from the transformations compared to the PCA of the neural activity themselves? The latter should reflect the functional neural network structure. I am curious if the transformation-based network is associated with the underlying functional network or not. Now it seems that as a data-driven map, it lacks a clear connection to what we already know about the language network.

This is a very interesting comment, and proved to be a good sanity check. To address this empirically, we applied the same PCA analysis directly to the time series of neural activity in the language network, rather than the headwise transformation weights learned by the encoding model. Below, we reconstruct the first two PCs from this analysis for visualization on the brain. We see that both PCs are fairly symmetric across hemispheres; this is in contrast to the transformation weight PCs in Fig. 4, which (especially in the case of PC2) appear to be more asymmetric. PC1 groups medial prefrontal, frontopolar, and posterior medial cortex together, differentiating these higher-level association cortices from lower-level sensory and language areas in lateral temporal and lateral prefrontal areas (similarly to Margulies et al., 2016; Samara et al., 2023). PC2 on the other hand hinges on posterior medial cortex, grouped with angular gyrus, and a familiar patch of lateral

temporal cortex—that is, the red-colored areas of PC2 seem to resemble the posterior hubs of the default mode network.

Critically, PCs 1 and 2 derived from the headwise transformation weights do not trivially reproduce these networks, and appear to map onto the brain in a more regionally-specific fashion. This is consistent with prior work on (within-subject) functional connectivity showing that the network structures recovered by this kind of PCA analysis will be dominated by intrinsic rather than stimulus-driven fluctuations (Simony et al., 2016). For the sake of space, we've opted not to include this analysis in the revised manuscript—but we're happy to include it if the reviewer sees fit. In keeping with Reviewer 2's comments, we have generally tried not to over-interpret the results in Fig. 4; however, we've added references to recent work on processing timescales to the Discussion in hopes of better contextualizing the results.

We also found that left-lateralized anterior temporal and anterior prefrontal cortices were associated with longer look-back attention distances (positive values along PC2; Fig. 4), suggesting that these regions may have longer temporal receptive windows (Lerner et al., 2011; Hasson et al., 2015; Chang et al., 2022; Vo, Jain et al., 2023) and compute longer-range contextual dependencies, including event- or narrative-level relations (Maguire et al., 1999; Vandenberghe et al., 2002; Ferstl et al., 2008; Makuuchi et al., 2009; Bašnáková et al., 2014; Baldassano et al., 2018).

6. The authors showed encoding results on 10 ROIs. How are these 10 ROIs picked? Are these all the ROIs that demonstrated significant speech-response?

We selected these ROIs to span the cortical hierarchy for language processing from low-level sensory areas (e.g. HG) to high-level association areas (e.g. PMC). All of these ROIs had reliable, stimulus-driven responses to the spoken stories as indexed by ISC (Fig. S1 in the original manuscript). We've added a paragraph to the "Experimental data" section of the Methods describing how we constructed the ROIs in more detail:

We constructed a set of 10 ROIs intended to span the cortical hierarchy for language and narrative processing from low-level sensory areas (e.g. HG) to high-level association areas (e.g. PMC). First, we extracted language ROIs from Fedorenko and colleagues (2010): PostTemp, AntTemp, AngG, IFG, IFGorb, and MFG. We roughly defined each ROI as the set of parcels from the 1000-parcel atlas in which over 50% of voxels in the parcel overlapped with voxels assigned to the ROI in MNI space. At the bottom of the hierarchy, we added early auditory cortex (Heschl's gyrus; HG), extracted from the Harvard-Oxford Atlas. At the upper end of the hierarchy, we added vmPFC, dmPFC, and PMC ROIs, manually defined from the Schaefer parcellation, to capture representation of higher-level narrative features and events (Baldassano et al., 2017, 2018). These ROIs encompass most of the brain areas with reliable, stimulus-driven activity when subjects listen to spoken stories (Nastase et al., 2021).

7. The computation of backward attention distance seems to be constrained within each layer. However the actual attention is cumulated over the layers, the current method may underestimate the actual attention distance, particularly for the later layers. (See Abnar & Zuidema 2020, Quantifying attention flow in transformers)

We agree that the computation of backward attention distance (Fig. 4E) is localized to each layer—this is a result of our focus on the transformations implemented by individual attention heads, which operate independently of other heads within a given layer. The reviewer makes an important point that attentional distances in fact accumulate over layers. In the "Generating Transformer features" section of the revised Methods, we have added a sentence pointing out that we may in fact be underestimating backward attention distances as they accumulate across layers:

Note that by focusing on backward attention distances for the transformations implemented by individual attention heads, we may underestimate attention distances that effectively accumulate over layers (Abnar & Zuidema, 2020).

We suspect that using a more sophisticated metric of attention flow would yield qualitatively similar results (and the manuscript is already pretty dense), so we're inclined to leave this question to future work.

8. The authors performed shuffling of the coefficients as a control analysis on Page 12. I wonder if the results would be consistent if using a randomized Transformer model as a control.

This is a great idea. We extracted transformations when supplying our stimuli to an untrained, randomly initialized instance of the BERT architecture, then re-ran the headwise correspondence analysis (Fig. 5). We found that the untrained model did not result in any obvious structure across dependencies or ROIs (see the figure below). This serves as a good control analysis, so we've included these new results in Fig. S28.

Figure S28. As a control analysis, we reevaluated headwise functional correspondence using an untrained, randomly-initialized instance of BERT. This is intended to be a strong control in that the model retains the same architecture as the trained model and receives the same text from our stimuli as input. The untrained BERT model, however, has not been trained to predict masked words across large corpora of text, and therefore the model's internal weights do not encode the statistical structure of real-world language. After extracting the transformation vectors from the untrained model based on our stimulus, we recomputed both the brain prediction scores (the encoding model mapping transformations onto parcelwise brain activity) and the dependency prediction scores (the logistic regression model for predicting the occurrence of a given linguistic dependency) for each ROI and each linguistic dependency. We then recomputed the functional correspondence of brain and dependency prediction scores across heads (cf. Fig. 5). The resulting correlations do not reveal any obvious structure across dependencies or ROIs. The untrained model (A) yields lower brain and dependency prediction scores and (B) reduces functional

correspondence (no significant correlations between brain and dependency prediction scores). **(C)**
The 95% bootstrap confidence intervals for the mean functional correspondence across dependencies for each ROI approach or cross zero, suggesting that there is no significant functional correspondence for any ROI.

Reviewer 2

Summary: While a majority of studies bridging large language models and neuroscience utilize the hidden states of the LMs as predictors of brain activity, this study proposes a novel framework to instead use the intermediate transformations (namely, the attention vectors in transformers). Their approach has the advantage that many studies in NLP have shown attention vectors to be highly specific and interpretable, especially for syntactic operations. The authors show that (1) these features predict as well as the hidden states (2) (due to the network architecture) exhibit more variability across layers than the hidden states (3) predict interesting gradients across the cortical surface with respect to layer selectivity and timescale.

Overall, this paper was clear, well motivated, interesting and carefully written so as to not over-claim. I believe that it brings a novel approach to bridge AI models and neuroscience. I only have minor questions and clarifications listed below.

We appreciate the reviewer's support for our approach, and have made an effort to address each comment below.

1. Why do the authors use group-averaging when they could have built encoding models for individual participants? Is it to facilitate the computation of ISC as a noise ceiling estimate?

To clarify, we did in fact build encoding models within each individual participant. We only aggregate the model performance scores (correlation between predicted and actual test time series) derived from the subject-specific models across participants for downstream statistical analysis and visualization. This is in contrast to iconic "low N " studies (e.g. Huth et al., 2016, $N = 7$), which similarly fit models within each subject, but statistically evaluate each subject separately.

We chose to statistically evaluate encoding performance across subjects (in line with other papers; e.g. Pereira et al., 2018; Toneva & Wehbe, 2019; Caucheteux et al., 2023) for several reasons. First, unlike the low N dataset used by Huth and colleagues (which was made publicly available in August of this year—nice! LeBel et al., 2023), the Narratives dataset comprises larger samples of subjects (e.g. $N = 45$ for the "I Knew You Were Black" story) with each subject exposed to fewer spoken story stimuli. Second, we estimated within-participant parcelwise encoding models based on the 1000-parcel atlas by Schaefer and colleagues (2018) to reduce computational demands across numerous models tested; we suspect that this spatial downsampling, while potentially reducing the topographic specificity of the encoding models to some degree (Fedorenko et al., 2010), may also provide coarse-grained alignment when aggregating within-subject model performance scores across subjects. Third, we estimate a between-subjects noise ceiling based on intersubject correlation (ISC; Nastase et al., 2019) for two reasons: (a) practically, we do not have multiple test sets for most participants in the Narratives dataset; and (b) estimating a within-subject noise ceiling based on repeated exposures to the same stimulus does not respect the novelty of the stimulus and will include memory and habituation processes in subsequent presentations (Aly et al., 2018; Michelmann et al., 2021).

We fully acknowledge that finer-grained encoding may be possible using voxelwise (rather than parcelwise) models, statistically evaluating each subject separately, or aggregating across subjects in a more sophisticated way (e.g. using hyperalignment or SRM; Van Uden, Nastase et al., 2017; Nastase et al., 2020)—but we had to make some compromises. In the “Statistical assessment” section in the Methods of the revised manuscript, we clarified the motivation for these compromises:

In all cases, parcelwise encoding models were estimated and evaluated within individual subjects. However, for statistical evaluation, we aggregate the model performance scores (correlation between predicted and actual test time series) derived from the subject-specific models across subjects. This approach is in contrast to “low N ” studies (e.g. Huth et al., 2016, $N = 7$), which similarly fit models within each subject, but statistically evaluate each subject separately. In our approach, finer-grained differences in cortical topographies for language across individual brains may be obscured by aggregating parcelwise results across subjects (Fedorenko et al., 2010; Braga et al., 2020). We compromised on this front for three reasons. First, in contrast to dense-sampling, low N datasets (e.g. LeBel et al., 2023), the Narratives dataset comprises larger samples of subjects (e.g. $N = 45$ for the “I Knew You Were Black” story) exposed to fewer spoken story stimuli. Second, we estimated parcelwise encoding models based on the 1000-parcel atlas by Schaefer and colleagues (2018) to reduce computational demands across numerous models tested; we suspect that this spatial downsampling, while reducing the specificity of the encoding models to some degree, may also provide coarse-grained alignment when aggregating within-subject model performance scores across subjects. Third, we do not have repeated exposures to the same test set (e.g. Huth et al., 2016), and therefore estimate a between-subjects noise ceiling based on intersubject correlation (ISC; Nastase et al., 2019). We acknowledge that finer-grained encoding may be possible by aggregating across subjects in a more sophisticated way (e.g. using hyperalignment; Van Uden, Nastase et al., 2017; Nastase et al., 2020).

2. It is interesting that the transformer magnitudes outperform GloVE embeddings in low-level areas. How correlated is this feature with word rate or part-of-speech tags? I wonder if its capturing periodicity across tokens (for example, the content words in English are distributed at specific positions) that matches areas like HG but is devoid of semantic content that would be relevant for AngG.

This is a great question. We hadn’t thought to examine how the transformation magnitudes track with lower-level features, but we agree with the reviewer that they are likely correlated. To address this question empirically, we first correlated the time series transformation magnitudes across TRs with word rate and phoneme rate. In our implementation, these are highly correlated: $r \approx .8$ in both cases. However, when fitting the encoding models for transformation magnitudes (and other feature sets), we included word rate and phoneme rate (as well as phoneme indicators) as confound variables in a separate band (to ensure they receive their own optimal hyperparameter). This suggests that although transformation magnitudes are in fact correlated with word rate and phoneme rate, they nonetheless capture some variance in temporal cortex above and beyond those lower-level features.

To examine whether the transformation magnitudes track with part of speech, we used least-squares regression to predict the time series of transformation magnitudes at each TR from the 14-dimensional part-of-speech indicator time series (the same parts of speech included in the “linguistic features” model used in Fig. 2). This yielded out-of-sample correlations on the scale of $r \approx .4$ across layers, suggesting that transformation magnitudes are also related to part of speech. This may explain why, although transformation magnitudes nominally outperform the linguistic features (including part of speech) in temporal ROIs, they perform comparably elsewhere.

First, we clarified in the revised text how these transformation magnitudes were computed and downsampled to the fMRI sampling rate:

Finally, to generate “transformation magnitudes” for each TR, we averaged the “transformation” vectors over all tokens in the TR, then computed the L2 norm of each attention head’s transformation vector.

We’ve added a sentence to the “Transformer-based features” section of the Methods to make the reviewer’s caveat explicit:

These transformation magnitudes may correlate with low-level features if particular word roles tend to be distributed at specific positions in natural speech. We found that the time series of transformation magnitudes are strongly correlated ($r \approx .8$) with word rate and phoneme rate in our spoken story stimuli. Note, however, that when fitting the encoding models for transformation magnitudes (and other feature sets), we included word rate and phoneme rate (as well as phoneme indicators) as confound variables in a separate band (to ensure they receive their own optimal hyperparameter). Interestingly, we also found that the transformation magnitudes are related to part of speech: we used least-squares regression to predict the time series of transformation magnitudes at each TR from the 14-dimensional part-of-speech indicator time series, resulting in a moderate out-of-sample correlation of $r \approx .4$.

3. Since the attention vectors are passed onto an MLP, the low correlation with the hidden state is to be expected. Similarly, the residual connection would cause higher autocorrelation in the hidden state. I think the authors should note this clearly in the text:

“Note that despite yielding similar encoding performance, the embeddings and transformations are fundamentally different; for example, the average TR-by-TR correlation between embeddings and transformations across both stimuli is effectively zero ($-.004 \pm .009$ SD), the embeddings and transformations yield visibly different TR-by-TR representational geometries (Fig. S4), and the transformations have considerably higher temporal autocorrelation than the embeddings (Fig. S5).”

Thanks for noting this. We’ve updated the relevant text to clarify that this should be expected given the architecture of the model.

Note that despite yielding similar encoding performance, the embeddings and transformations are fundamentally different; for example, the average TR-by-TR correlation between embeddings and transformations across both stimuli is effectively zero ($-.004 \pm .009$ SD), **the embeddings and transformations are not correlated across layers (Fig. S5)**. This is expected, given that embeddings and transformations reside in different feature spaces, where the transformations are translated into the embedding space by an MLP. The embeddings and transformations also yield visibly different TR-by-TR representational geometries (Fig. S6), and the transformations have considerably higher temporal autocorrelation than the embeddings (Fig. S7).

Based on a comment from Reviewer 1, we've also several paragraphs to the Methods to more explicitly describing the relationships between the transformations and embeddings from layer to layer (reproduced here for convenience):

Intuitively, it may seem as if the transformations are to some extent redundant with the embedding at the previous layer, or the resulting embedding passed to the subsequent layer. The transformations in layer x are not computed from the embedding at layer $x - 1$ in a particularly direct or straightforward way. Rather, the transformations at layer x are the result of the complex interplay between the key-query-value ($k-q-v$) vectors, which are themselves a function of the embedding at layer $x - 1$. The learned weights at each attention head specify a projection from the embedding at layer $x - 1$ to a set of $k-q-v$ components, which in turn determine a nonlinear function for pulling in and combining contextual information from other tokens. Thus, although the resulting transformations at layer x share the same dimensionality with the embedding at $x - 1$, they encode fundamentally different kinds of information.

The embedding is *cumulative* – it carries both the not-yet-contextualized semantic content from layer 0 (the initial token embedding), as well as a linear combination of other contextual information incorporated by transformations at prior layers. The transformations can instead be thought of as encoding context-appropriate “adjustments” or “diffs”. These adjustments are added linearly into the embedding passed along from the previous layer, effectively sculpting the embedding to respect the context. In fact, the embedding at layer $x - 1$ can pass through the attention heads largely unchanged via the so-called “residual stream”; the model learns when and how each transformation should adjust the embedding based on context (Elhage et al., 2021).

These adjustments are added to the embedding, effectively sculpting the embedding to respect the context. The transformations are not natively “aligned” with the embedding; they are passed through another nonlinear transformation—the MLP—that translates the transformations into the embedding space in order to add them to the embedding at layer x . This step effectively fuses the contextual information derived from other words with the content of the current word embedding. Thus, the adjustments implemented by the transformations are ostensibly “contained” in the new embedding at layer x , but they are nonlinearly fused with the content of the previous layer.

To specifically highlight that the residual connections will cause higher autocorrelation across layers, we've modified the text in the Results and Fig. S9 caption:

We next segregated the Transformer features into separate layers. There is an important theoretical distinction in the layer-by-layer structure of the embeddings and transformations arising from the architecture of the network. The embeddings encode the meaning of the current word and become increasingly contextualized from layer to layer (Tenney et al., 2019). Residual connections allow the embeddings to propagate and accumulate information across layers (Elhage et al., 2021). The transformations, on the other hand, capture the “updates” to the embedding at each layer—derived from other words in the surrounding context. The transformations are largely independent from layer to layer (Fig. S9) and produce more layer-specific representational geometries (Figs. S10, S11). Based on these distinct computational roles, we hypothesized that the transformations would map onto the brain in a more layer-specific way than the embeddings.

Fig. S9... Embeddings are much more similar across layers because the residual connections allow the embeddings to accumulate information (or remain unchanged) across layers. On the other hand, transformations capture layer-by-layer “updates” to the embedding and are largely uncorrelated across layers.

4. Re arguing that they are “fundamentally different”: Obviously, the two representations are related to each other. But the MLP & residual connections imply that they don’t encode information the same way. Moreover, BERTology has shown that while it is difficult to understand information encoded in the hidden states, individual attention heads are often interpretable + Your results show very interesting differences in encoding performance and layer hierarchy!

We appreciate the reviewer’s enthusiasm about our approach! We also hope that our reply to Reviewer 1’s comment (reproduced in the comment immediately above) will clarify some of these relationships.

5. I agree with the arguments presented here to support the claim that “suggest that the computations implemented by the transformations are more layer-specific than the embeddings”, namely (1) the known function of different attention heads in the literature and (2) the residual connections across hidden states which would make them more similar (it would be nice to note this explicitly). However, isn’t a stronger form of evidence here showing the amount of unique vs. shared variance explained by hidden states/attention vectors across different layers?

This is a great suggestion, and Reviewer 1 also keyed into a similar idea. As a straightforward way to assess unique variance explained by the embeddings and transformations, we computed partial correlations between the model-based predictions derived from the embeddings and transformations against the actual brain activity (see the figure below). This revealed that, across the cortical language network, the transformation-based predictions capture more unique variance than the embedding-based predictions at early layers of the model. The embedding-based predictions capture more unique variance at later layers of the model. This suggestion turned out to provide a

compelling view onto one of our core results, so we've incorporated this analysis into the main text and visualize the results as a new panel in Fig. 3.

Figure 3. (B) Partial correlations between brain activity and model-based predictions derived from embeddings (blue) and transformations (red). For each layer, we measured the correlation between transformation-based predictions and brain activity while controlling for the embedding-based predictions (and vice versa). Partial correlations at each layer were averaged across parcels in the cortical language network. Error bars denote 95% bootstrap confidence intervals across subjects

We also added the following passage the Results section:

To evaluate the unique contributions of transformations- and embedding-based predictions at each layer, we performed a partial correlation analysis: we measured the correlation between transformation-based predictions and brain activity while controlling for the embedding-based predictions (and vice versa; Fig. 3B). We found that Transformation-based predictions capture more unique variance at earlier layers than embedding-based predictions; embeddings, on the other hand, accumulate information over time and capture the most unique variance at later layers.

In the reply to the previous comment (reproduced here for clarity), we've also explicitly noted that we expect the embeddings to be more similar across layers due to the residual connections:

Residual connections allow the embeddings to propagate and accumulate information across layers (Elhage et al., 2021).

Fig. S9... Embeddings are much more similar across layers because the residual connections allow the embeddings to accumulate information (or remain unchanged) across layers. On the other hand, transformations capture layer-by-layer “updates” to the embedding and are largely uncorrelated across layers.

5. Minor: For the analysis in Fig. 4, were the same regularization coefficients used for the different encoding models as that could give rise to arbitrary differences in scale across parcels and participants.

When fitting the transformations-based encoding model that we ultimately analyzed for Fig. 4, all transformations were fit within a single band during ridge regression. This means that, for a given parcel, the coefficients for different transformations should be on a similar scale. To account for the fact that the learned weights may be on different scales at different parcels (due to different regularization parameters), we z-scored the weight vectors for each parcel prior to subsequent analysis. We note this explicitly in the “Summarizing headwise transformation weights” in the revised Methods text:

To account for the fact that the learned weights may be on different scales at different parcels (due to different regularization parameters), we first z-scored the weight vectors for each parcel prior to subsequent analysis.

6. I appreciate that the authors didn't overstate the results in Fig. 4-5. However, I am curious to hear how these results can be placed in the context of previous work like Huth et al., 2016 and Lerner et al., 2011 that also make predictions about cortical gradients like in Fig. 4. Since the relationship between function and ROI was based on merely correlating the dependency prediction with the encoding model weights, couldn't it be possible that these regions don't have a purely syntactic role but their function is related to this property? (I know that the authors never claimed this, just curious to know what they think) Specifically, can we expect any differences in functional predictions made by this approach based on whether an ROI is correlated with many syntactic markers like AngG/MFG vs. ROIs that don't correlate with any like vmPFC? How does this fit with more recent studies that show no dissociation between syntax and semantics?

We appreciate the reviewer's understanding here—it's been difficult to strike the right balance between not over-interpreting the results of Figs. 4 and 5 while nonetheless trying to situate them with respect to the literature. The gradients depicted in Fig. 4 are most easily understood as summarizing the language network's top two (orthogonal) tuning curves across the transformations implemented by 144 attention heads in BERT. These transformations are not strictly syntactic, as we discuss elsewhere in the manuscript; rather, they encode a variety of different operations for incorporating contextual information from other words into the current word. Headwise transformations with higher weights contribute more strongly (positively or negatively) to predicting brain activity in certain parts of the language network. We plot model features like *layer* and *backward attention distance* along these PCs in hopes of more easily interpreting these tuning curves.

In the case of PC1, positive values corresponding to earlier layers of the model are strongly localized to posterior temporal areas, whereas intermediate and negative values corresponding to intermediate and later levels of the model are distributed throughout the language network. For PC2, positive

values correspond to longer lookback distances. Interestingly, these long lookback distances are somewhat left-lateralized and map onto anterior temporal and anterior prefrontal areas. The reviewer is correct that these results resonate with work on temporal receptive windows. We've made minor updates to paragraph 4 of the Discussion in hopes of clarifying the meaning of these results, and added references to Lerner et al., 2011, and a recent, related study by Vo, Jain et al., 2023:

We also found that left-lateralized anterior temporal and anterior prefrontal cortices were associated with longer look-back attention distances (positive values along PC2; Fig. 4), suggesting that these regions may have longer temporal receptive windows (Lerner et al., 2011; Vo, Jain et al., 2023) and compute longer-range contextual dependencies, including event- or narrative-level relations (Maguire et al., 1999; Vandenberghe et al., 2002; Ferstl et al., 2008; Makuuchi et al., 2009; Bašnáková et al., 2014; Baldassano et al., 2018).

High correlation values in our headwise functional correspondence analysis (Fig. 5B) indicate the following relationship: headwise transformations that more robustly encode information about a given linguistic dependency also tend to better predict activity in a given ROI. We interpret this to mean that ROI encodes features of the stimulus that particular transformations of the model also preferentially encode. Low correlation values, as we observe when shuffling transformation features across heads (Fig. S27) or with the new analysis suggested by Reviewer 1 using an untrained BERT architecture (Fig. S28), indicate no systematic relationship between which heads best predict a given dependency and which heads best predict activity in a given ROI.

For ROIs with high correspondence scores across multiple dependencies, like MFG and AngG, this indicates that heads that encode a variety of different dependencies also predict brain activity well. This would suggest that these ROIs encode certain syntactic and/or contextual features of the stimulus (because the transformations incorporate information across words) in a way that matches the headwise functional specialization of the model. However, we want to be cautious not to over-interpret this result, given that this effect may be driven in part by a shared subset of heads with strong predictions across several dependencies.

We agree with what the reviewer is implying in this comment: we think it's unlikely that any cortical region is engaged in purely syntactic computations; we suspect that cortical processing of syntax is likely entangled with other features like semantics (Fedorenko et al., 2020; Caucheteux et al., 2021). There is no reason for a neural network (be it a model or the human brain) to structurally factorize or modularize syntax relative to other linguistic features, unless this is necessary to fulfill the network's objective function (Hasson et al., 2020). Although the objectives that have shaped the human language system are not well understood, large language models suggest that relatively simple objective functions, such next-word prediction (especially combined with social feedback in more modern models), are sufficient to process natural language and produce syntactically sound linguistic outputs (albeit based on an inhumanly large volume of training data; Piantadosi, 2023). From an analytic perspective, the reviewer is correct that we cannot infer that a model is "correct" just based on modest encoding performance:

Although we believe these parallels position LLMs as useful models of human language processing, we caution that just because a model predicts brain activity better than other models does not make it a “good” model (Schaeffer et al., 2022; Antonello & Huth, 2023; Guest & Martin, 2023).

Reviewer 3

1. The manuscript is well-written and the figures are pleasant to follow – the topic is potentially a new take on what aspects of transformer architectures to correlate with neuroimaging data. I wanted to be excited about this work but I ended up being baffled by its motivation and meaning, and later frustrated by the weak analytical argumentation (lack of comparisons/controls supporting the conclusions). The manuscript was also peppered with unsubstantiated claims (specifically what transformers learn, and if they learn linguistic structure – that has by no means been rigorously tested and convincingly shown, yet is discussed as if it is given and not a contentious point of ongoing debate).

We appreciate the thoughtful comments, and are especially grateful to the reviewer for helping to pinpoint where our controls may have fallen short and where our claims may have been overstated. As you will see below, we have implemented several new controls (and strengthened our existing controls) as suggested. We have also put a good deal of effort into better articulating our theoretical commitments and hypotheses. If in the revised manuscript any of the reviewer’s concerns remain unaddressed, we are happy to make targeted adjustments where the reviewer sees fit.

To take one example, we have carefully reined in any claims as to “what transformers learn”, and have provided more comprehensive references to support our newly limited claims. We did not intend to take a strong stance on whether transformers learn any underlying “rules” of linguistic structure (let alone the semantic “meaning” of language, which in our understanding has been the most contentious point). We only meant to refer to the empirical claim that they increasingly approximate human-like patterns of syntactic generalization as measured by benchmarks such as BLiMP (Warstadt et al., 2020; see Linzen & Baroni, 2021, and Mahowald et al., 2023, for reviews). There will certainly be newer and more human-like models that continue to emerge as research progresses. Our aim is *not* to crown any particular model as the most brain-like, but to expand the area’s analytic paradigm from an over-emphasis on the embeddings of these models to include the more functionally specialized internal transformations that produce them.

Most mystifyingly, the manuscript ignores the large body of work by the Brennan and Hale labs that relate parser states to brain states. That was very puzzling and seemingly a huge oversight – that work ^{is} the literature on relating NLP models to neural data that transformer states should be compared to. See for example: Stanojević, M., Brennan, J. R., Dunagan, D., Steedman, M., & Hale, J. T. (2023). Modeling Structure-Building in the Brain With CCG Parsing and Large Language Models. *Cognitive Science*, 47(7), e13312.

We are grateful that the reviewer highlighted this rich line of work from Jonathan Brennan’s and John Hale’s labs—we had unfortunately overlooked it in our focus on transformer architectures. We appreciate their positive reference to our current preprint in the Stanojević et al (2023) paper: “Kumar et al. propose an interesting strategy to confront this [“black box”] limitation by focusing specifically on how different “attention heads” in the transformer network, which serve in this architecture to mediate the spread of feed-forward activation as a function of context, drive neural performance in different brain regions. Analyzing the distribution of activation across attention heads is one of

several strategies that have been pursued with some success in rendering LLMs interpretable in terms of linguistic structure.” We have now included more thorough references to Brennan’s and Hale’s work throughout the manuscript.

We added the following passage to the Introduction:

This is in contrast to probabilistic syntactic parsers (e.g. Dyer et al., 2016; Hale et al., 2018; Brennan et al., 2020), which learn to reproduce a predefined set of syntactic labels to construct parse trees. The transformations do not explicitly disentangle syntax from the meaning of words and do not rely on predefined labels; instead they learn to approximate whatever contextual structures are useful for accurately predicting words in real-world text.

We’ve added the citations to the following work from Brennan and Hale to the manuscript:

Brennan, J., Nir, Y., Hasson, U., Malach, R., Heeger, D. J., & Pylkkänen, L. (2012). Syntactic structure building in the anterior temporal lobe during natural story listening. *Brain and Language*, 120(2), 163–173. <https://doi.org/10.1016/j.bandl.2010.04.002>

Brennan, J. (2016). Naturalistic sentence comprehension in the brain. *Language and Linguistics Compass*, 10(7), 299–313. <https://doi.org/10.1111/lnc3.12198>

Brennan, J. R., Dyer, C., Kuncoro, A., & Hale, J. T. (2020). Localizing syntactic predictions using recurrent neural network grammars. *Neuropsychologia*, 146, 107479.

Dyer, C., Kuncoro, A., Ballesteros, M., & Smith, N. A. (2016). Recurrent neural network grammars. In Knight, K., Nenkova, A., & Rambow, O. (Eds.) *Proceedings of the 2016 Conference of the North American Chapter of the Association for Computational Linguistics: Human Language Technologies* (pp. 199–209). <https://doi.org/10.18653/v1/N16-1024>

Hale, J. T., Campanelli, L., Li, J., Bhattasali, S., Pallier, C., & Brennan, J. R. (2022). Neurocomputational models of language processing. *Annual Review of Linguistics*, 8, 427–446. <https://doi.org/10.1146/annurev-linguistics-051421-020803>

Stanojević, M., Brennan, J. R., Dunagan, D., Steedman, M., & Hale, J. T. (2023). Modeling structure-building in the brain with CCG parsing and large language models. *Cognitive Science*, 47(7), e13312. <https://doi.org/10.1111/cogs.13312>

We read with great interest about the combinatory categorial grammar (CCG) used by Stanojević and colleagues (2023) to capture syntactic operations in sentence processing. In line with the reviewer’s comment, we used the open CCG code and examples provided by Stanojević and colleagues (<https://github.com/stanojevic/ccgtools>—yay, open code!) to parse the transcript of our stimulus and extract their metric of parsing effort. Our code for applying the CCG model can be found here. We extracted the scalar parsing effort metric for right-branching, left-branching, and left-branching-with-revealing operations, resulting in three effort scores for each word in the transcript. We summed each score across words within a given TR to downsample the word-level effort scores to the temporal resolution of the fMRI acquisition. We then supplied the three CCG

scores to the same encoding pipeline estimated using banded ridge regression and evaluated using three-fold cross-validation on held-out data.

The CCG effort metric yielded notable prediction scores in early auditory cortex, posterior and anterior temporal cortex, as well as angular gyrus, corroborating the results reported by Stanojević and colleagues (2023). In early auditory cortex, the CCG effort metric performed comparably with the embeddings and transformations, but was outperformed by the transformation magnitudes. In all other ROIs, however, the transformer features outperformed CCG effort. For example, outside of early auditory cortex, CCG effort performed best in anterior temporal cortex—but the BERT transformations performed over three times better in this ROI.

We've included this comparison in a new supplementary Fig. S3:

Figure S3. Comparing combinatory categorial grammar (CCG) parser effort to other linguistic features. Note that only the “I Knew You Were Black” story stimulus was used in this comparison. Scalar parsing effort metrics for right-branching, left-branching, and left-branching-with-revealing operations were supplied to the same encoding pipeline used for other types of features: banded ridge regression with three-fold cross-validation. The CCG effort metric (brown) yielded above-zero prediction scores in early auditory cortex, posterior and anterior temporal cortex, as well as angular gyrus, corroborating the results reported by Stanojević and colleagues (2023). In early auditory cortex, the CCG effort metric performed comparably with the embeddings and transformations, but was outperformed by the transformation magnitudes. In all other ROIs, however, the transformer features outperformed CCG effort.

We added a sentence to the revised Results section pointing to this result.

Embeddings and transformations also outperform an “effort” metric extracted from a state-of-the-art incremental sentence parser (Stanojević et al., 2023) in almost all language ROIs (Fig. S3).

We also added a paragraph to the “Baseline linguistic features” of the revised Methods to describe how these features were extracted:

We also used a modern combinatory categorial grammar (CCG) to capture symbolic syntactic operations in sentence processing (Stanojević et al., 2023). CCGs are incremental parsers that explicitly model syntactic structure, but with more human-like expressiveness than widely-used context-free grammars (CFGs). We used code and examples provided by Stanojević and colleagues (<https://github.com/stanojevic/ccgtools>) to parse the transcript of the “I Knew You Were Black” stimulus and extract their metric of parsing effort. We extracted the scalar parsing effort metric for right-branching, left-branching, and left-branching-with-revealing operations, resulting in three effort scores for each word in the transcript. We summed each score across words within a given TR to downsample the word-level effort scores to the temporal resolution of the fMRI acquisition.

We note that evaluating the scalar metric of CCG parsing effort against our LLM features may underestimate the information made available from the parser. To more equitably compare these different families of models, we may need to vectorize both the word embeddings and internal parsing operations—e.g. shift, reduce, reveal—from the CCG model. According to a note by Stanojević and colleagues (2023, p. 33), “The parser has 425 different shift transitions, one for each possible lexical category, and 20 different reduce transitions (5 for binary combinators, 2 for unary type-raising combinators, 12 for unary type-changing rules and 1 for the revealing operation).” This suggests that it may be possible to fully vectorize the internal structure of the parser into a 400+-dimensional model for an encoding analysis. However, we are not aware of any examples where the authors perform an analysis of this kind, and do not consider ourselves qualified to “unpack” the internal workings of their CCG implementation in this way. We hope that in future work we can collaborate to consider these possibilities.

2. At first I was very interested in this new approach of “transformations” but was ultimately disappointed in the lack of theoretical grounding for the idea of transformations. Without theory, the work is very well-executed engineering exercise – namely, the big worry here, what is the theory and the hypothesis being tested here? If the proposal is that transformers and brains are doing something “the same” or “alike,” then what, specifically, is that thing? Is the brain doing what a transformer is doing, if so, how? There are so many problems here (e.g., the brain has time, transformers don’t, copious behavioral and formal capacities, etc., nevermind training data and training protocol differences). This quote from the Abstract, might be a good way to highlight the problem: “LLM and the cortical language network may converge on similar trends of functional specialization for processing natural language”- ok, but what are those “similar trends”? How could they arise given the stunning and fundamental differences between the two systems? What would “similar trends” mean?

Then, why are some parts of the transformer architecture (here, transformations and the brain regions) better fits than other parts (embeddings)? If you're finding that some system with regularities has regularities that are more similar to another system with regularities, you're really just saying there are correlations between regularities (which is trivially true), unless you're interpreting those regularities to be meaningful (e.g., the two systems are doing the same thing, behaving the same way, or whatever) – for that you need a theory and a hypothesis about what is similar between the systems, why it is so, and what it means.

Relatedly, if the conclusion is, transformer states predict brain states ergo transformers and brains are doing something similar, there are two problems. First, on a high level that is trivially true – they are both processing sequential language stimuli – although with very different behaviors, outcomes, physical and formal expressive constraints. But logically, the argument falls apart; we see this in the history of science with epicycles and geocentrism – to cut a long story short, basically, geocentric models could keep adding parameters to better predict the movement of celestial bodies than heliocentric models could. Geocentric models predicted better, but the sun is at the center of our solar system. Second, given that not all aspects of the transformer predict neural data, and not all areas of the brain are predicted, why is the transformer's failure to predict not interpreted as “doing something fundamentally different”? These two logical pitfalls have recently been discussed at length here:

Guest, O., & Martin, A. E. (2023). On logical inference over brains, behaviour, and artificial neural networks. *Computational Brain & Behavior*, 1-15.

It is hard to evaluate whether transformations are an improvement, whether that improvement has meaning or not, when they are not compared to performance by parsers (e.g., a la Brennan and Hale work, RNN-G). Another option would be to illustrate through comparison with equally sparse predictors as the “classical linguistic annotations”. For the record, POS is not a feature that in and of itself is typically used to predict neural states, so it seems unfair to use that as a benchmark. If the goal of the work is to show that transformations are better predictors than embeddings, that seems more of an engineering exercise. It could be strengthened into a theory by theorizing as suggested above, and then by careful model evaluation across classes of existing successful models. Now as it stands the meaning of this work is frustratingly unclear.

We appreciate the reviewer pushing us to clarify our theoretical commitments in examining whether and how the internal computations of a transformer-based large language model relate to human brain activity. We hope to productively engage with an already large volume of work from a variety of groups suggesting that there are certain similarities between how LLMs and the human brain process language (e.g. Toneva & Wehbe, 2019; Schrimpf et al., 2021; Caucheteux et al., 2022; Goldstein et al., 2022; Heilbron et al., 2022). Our initial theoretical interest in isolating the “transformations” came from a critical perspective on this proliferation of work: all of these high-profile publications focus on the embeddings of the language model, but otherwise ignore the internal computations of the model. Our principal theoretical interest in the transformations—that is, the vector comprising the headwise output of the self-attention mechanism at each layer—lies in the fact that these are the components of the model that introduce contextual information extracted from prior words into the current word. In these models, any syntactic or compositional structure linking

one word to another must emerge from the transformations implemented by the attention heads. Without these transformations, the model effectively reduces to static, non-contextual lexical-semantic embeddings, like those learned by word2vec (Mikolov et al., 2013). The architecture of the transformer allows us to isolate the transformations implemented by individual attention heads, which have been shown to exhibit some degree of functional specialization for linguistic operations (e.g. Clark et al., 2019; Tenney et al., 2019).

We've added the following passage and subsequent paragraph to the Introduction:

These transformations are of particular theoretical interest, because they are the unique component of the circuit that allows information to flow between words: whatever syntactic or contextual information impacts the meaning of the current word is introduced solely via the transformations.

In the current work, we argue that the headwise transformations—the functionally-specialized contextual computations implemented by individual attention heads—can provide a novel window onto linguistic processing in the brain (Fig. 1A). A neurocomputational theory of natural language processing must ultimately specify how meaning is constructed across words. The Transformer architecture provides explicit access to a candidate mechanism for quantifying how the meaning of past words are incorporated into the meaning of the current word. If this is an important part of human language processing, these transformations should provide a good basis for modeling human brain activity during natural language comprehension. We extract transformations from the widely-studied BERT model (Devlin et al., 2019; Rogers et al., 2020) and use encoding models to evaluate these transformations against several other families of linguistic features in terms of predicting brain activity during natural language comprehension (Fig. 1B, C). We find that the transformations perform comparably to the embeddings, and generally outperform both non-contextual embeddings and classical syntactic annotations—suggesting that the contextual information extracted from surrounding words is surprisingly rich. In fact, transformations at earlier layers of the model account for more unique variance in brain activity than the embeddings themselves. Finally, we disassemble these transformations into the functionally specialized computations performed by individual attention heads. We find that certain properties of the heads, such as look-back distance, dominate the mapping between headwise transformations and cortical language areas. We also find that, for some language regions, headwise transformations that preferentially encode certain linguistic dependencies also better predict brain activity.

We've added two paragraphs to the beginning of the Results section:

We adopted a model-based encoding framework (Naselaris et al., 2011; Yamins & DiCarlo, 2016; Richards et al., 2019) in order to map Transformer features onto brain activity measured using fMRI while subjects listened to naturalistic spoken stories (Fig. 1A). Our principal theoretical interest lies in the transformations, because these are the components of the model that introduce contextual information extracted from other words into the current word. In these models, any syntactic or compositional structure linking one word to another must emerge from the transformations implemented by the attention heads. Although the transformations may approximate certain

syntactic operations, they do not explicitly disentangle syntax from the meaning of words and can incorporate content-rich contextual relationships. Given that the cortical language network also does not appear to cleanly differentiate syntax and other linguistic features (Fedorenko et al., 2012, 2020; Mesulam et al., 2015; Blank et al., 2016; Reddy & Wehbe, 2020; Caucheteux et al., 2021c), we theorized that the transformations may provide a useful basis for modeling neural activity during natural language processing.

We pursued two core questions: First, what is the efficacy of these transformations in predicting brain activity relative to both embeddings and other types of language features? We hypothesized that (a) the transformations would predict brain activity better than other types of language features; and (b) that the transformations would map onto cortical language areas in a more layer-specific way than embeddings, given that the embeddings accumulate contextual information across layers. Second, we address the exploratory question of whether the functional specialization observed in the transformations implemented by individual attention heads maps onto brain activity in a structured way. We operationalize “shared functional specialization” as a correspondence wherein headwise transformations that preferentially encode linguistic dependencies also better predict brain activity.

We’ve also added shorter passages at each stage of the Results in hopes of clarifying our hypotheses and the logic of our control analyses.

In the section titled “Transformer-based features outperform other linguistic features”, we added the following theoretical motivation:

The transformations are the conduit by which syntactic and contextual information are incorporated into the current word. We hypothesized that this rich contextual information would put the transformations on par with the embeddings in terms of predicting brain activity, and that the transformations would outperform other linguistic features.

In the section titled “Layerwise performance of embeddings and transformations”, we added the following paragraph:

We next segregated the Transformer features into separate layers. There is an important theoretical distinction in the layer-by-layer structure of the embeddings and transformations arising from the architecture of the network. The embeddings encode the meaning of the current word and become increasingly contextualized from layer to layer (Tenney et al., 2019). Residual connections allow the embeddings to propagate and accumulate information across layers (Elhage et al., 2021). The transformations, on the other hand, capture the “updates” to the embedding at each layer—derived from other words in the surrounding context. The transformations are largely independent from layer to layer (Fig. S9) and produce more layer-specific representational geometries (Figs. S10, S11). Based on these distinct computational roles, we hypothesized that the transformations would map onto the brain in a more layer-specific way than the embeddings.

We added the following passages clarifying the interpretation of the control analyses to the section “Interpreting transformations via headwise analysis”:

We next devised a control analysis to test whether the structure observed in Fig. 4 depends on the organization of transformation features into functionally specialized heads.

This control analysis indicates that the structure observed in Fig. 4 **does not arise trivially, and** results from the grouping of transformation features into functionally specialized heads; transformation features map onto brain activity in a way that systematically varies head by head, and shuffling features across heads (even within layers) disrupts this structure (Fig. S24).

To ensure the observed correspondence does not arise trivially, we designed two control analyses. In the first control analysis, we shuffled the transformation features across heads within each layer of BERT and then performed the same functional correspondence analysis. **This control analysis tests whether the observed correspondence depends on the functional organization of transformation features into particular heads.** Perturbing the functional grouping of transformation features into heads reduced both brain and dependency prediction performance and effectively abolished the headwise correspondence between dependencies and language ROIs (Fig. S27). **In the second control, we supplied our stimulus transcripts to an untrained, randomly-initialized BERT architecture, extracted the resulting transformations, and evaluated headwise correspondence with the brain. Headwise functional correspondence was similarly abolished for the untrained model (Fig. S28).** This indicates that the correspondence is not simply a byproduct of the model's architecture or our experimental stimuli, but depends in part on the model learning certain statistical structures in real-world language.

We also made heavy modifications to the Discussion section in an effort to both reiterate our theoretical motivation and clarify the meaning of our findings:

In the current study, we evaluated the transformations implemented by the attention heads of BERT against several other kinds of linguistic features in terms of predicting brain activity. There are three properties of the transformations that motivated our analyses. First, the transformations are the component of the model that allow information to flow between words: all syntactic, compositional, and contextual relations *across* words are generated by the transformations. Unlike classical syntactic annotations and syntactic parsers, however, the transformations do not explicitly disentangle syntax from word meaning and will learn to approximate whatever contextual relations are useful for predicting words in real-world text. Second, unlike the embeddings, which accumulate information from layer to layer, the transformations encode “updates” to the embedding at each layer, derived from the surrounding context. Third, the transformations at a given layer can be disassembled into the functionally specialized computations performed by individual attention heads. These properties are of particular theoretical interest for understanding how neural systems like the brain construct context-rich meaning across individual words in natural language.

We found that the transformations perform on par with the embeddings and outperform other linguistic features across most language ROIs, suggesting that the contextual information the transformations extract from surrounding words is surprisingly rich. We also found that the transformations at earlier layers of the model account for more unique variance than the

embeddings, and map onto cortical language areas in a more layer-specific fashion. Examining the contribution of headwise transformations to encoding performance reveals gradients in a low-dimensional cortical space that reflect certain structural and functional properties of the headwise transformations, including layer and look-back distance. Finally, we quantified the correspondence between headwise predictions of brain activity and syntactic dependencies for a variety of cortical language areas and dependencies, and found that headwise transformations that best predict certain dependencies also best predict certain ROIs (e.g. PostTemp, AngG, MFG). We show that this correspondence does not arise arbitrarily, but depends on the functional grouping of transformations into heads, and on the model’s architecture and training regime.

We added the following passage to the Discussion better articulate our theoretical interest in the transformations:

Indeed, much of our theoretical interest in the transformations stems from the observation that, although they approximate syntactic operations to some extent, they can also more expressively code for content- and context-rich relationships across words. We attribute the relatively strong prediction performance of the transformations to this rich contextual information.

We’re sympathetic to efforts to mediate the hype surrounding LLMs; we’ve tried to clarify some broader theoretical commitments and caveats in the revised Discussion:

Although the transformations yield relatively strong prediction performance, they are not positioned to serve as mechanistic models of cortical language processing—just because a model predicts brain activity better than other models does not make it a good model (Schaeffer et al., 2022; Antonello & Huth, 2023; Guest & Martin, 2023). That said, the transformations occupy a relatively underexplored space among theories of language processing: the transformations capture content-rich contextual relationships across words that inflect the meaning of the current word. This is in contrast to lexical-semantic models that capture only the meaning of individual words and parsing models that capture syntactic relationships among words without contextual meaning. While the embeddings (and resulting word predictions) are the “final product” of the contextualization process in Transformer-based models, the transformations capture the contextual processes that build up to this end point. More broadly, the human language system and end-to-end deep language models do—to some extent—share common computational principles aimed at producing well-formed, context-sensitive linguistic outputs (Hasson et al., 2020; Goldstein, Zada, et al., 2022): these models represent the unique meaning of words in context; these models are not “given” linguistic structures, but rather learn linguistic structure from natural language using a biologically-plausible, self-supervised objective function; these structures are encoded in high-dimensional embedding spaces across relatively simple computing elements; finally, these models better reproduce human-like performance than prior generations of models across a variety of natural language tasks (e.g. Devlin et al., 2018; Wang et al., 2018, 2019; Radford et al., 2019; Warstadt et al., 2020; Mahowald et al., 2023).

Finally, we note that our “classical linguistic features” do not just include part of speech; they include 14 part of speech indicators alongside 25 linguistic dependency indicators returned by spaCy; as

described in the “Baseline language features” section of the Methods. These classical linguistic indicators perform comparably to, or in some cases outperform, the “effort” metric extracted from the CCG parser introduced by Stanojević and colleagues (2023) across all ROIs excluding early auditory cortex.

Minor comments

3. What is the “neurobiology of language in silico”? Is this the claim that you can study the brain by using transformers? If so, what is the argument for why you can? Correlation?

We used this term to indicate efforts to leverage neurally-inspired computational models to better understand human language processing, in line with other recent papers (e.g. Jain et al., 2023; Vo et al., 2023). We hope to have clarified our position as to how we evaluate computational models against brain activity in the response to the reviewer’s prior comments. In any case, we agree with the reviewer that this terminology may be overly provocative, and have removed it:

Recently, a new class of artificial neural networks based on the Transformer architecture has revolutionized the field of language modeling, **attracting attention from neuroscientists seeking new techniques to understand the neurobiology of language.**

4. Time is totally different in the brain and in transformers’ transformation (spoiler: there isn’t any) – how were the number of TRs chosen? It seems arbitrary at the moment.

The reviewer is correct that most transformer architectures—including the two we focus on here, BERT and GPT-2—have no built-in representation of time. They operate on sequences of words or tokens, which in natural language unfold over time in sentences and discourse, but they do not respect subword timing or prosodic timing across words. Some newer transformer-based speech recognition models, such as Whisper (Radford et al., 2022), operate over continuous acoustic speech signals. These models have been shown to approximate phoneme structures (Goldstein et al., 2023) and may encode other subword and prosodic temporal features. In line with recent work (e.g. Vaidya et al., 2022; Défossez et al., 2023), we’re looking forward to exploring whether models that better respect the finer-grained temporal characteristics of speech can provide novel insights into human brain activity. We made a minor modification to the Discussion to make this limitation more explicit:

Second, the current work sidesteps the acoustic and prosodic features of natural speech (Santoro et al., 2014; de Heer et al., 2017); **the models we used operate on sequences of tokens in text and do not encode finer-grained temporal features of speech.** Future work, however, may benefit from models that extract high-level contextual semantic content directly from the **temporally-resolved** speech signal (in the same way that CNNs operate directly on pixel values; Li et al., 2022; Millet et al., 2022; Vaidya et al., 2022; **Goldstein et al., 2023**).

In the current work, the number of TRs was determined by the duration of the naturalistic spoken story stimuli; e.g. the “I Knew You Were Black” story told live by Carol Daniel for the Moth Radio Hour (<https://themoth.org/stories/i-knew-you-were-black>) was 800 seconds (13 minutes, 20 seconds) long, corresponding to 534 TRs with a 1.5-second TR. With fMRI, we cannot robustly resolve timing within individual words in real-world speech; most TRs contain multiple words and the BOLD response itself is temporally smoothed across several seconds. We adopt the standard procedure of concatenating the encoding features at multiple lags prior to fitting the encoding model (e.g. Huth et al., 2016). This finite impulse response (FIR) model accounts for varying hemodynamic lags across parcels and subjects by predicting each time point from a linear combination of features across varying lags. We note that prior work has shown that fMRI can access dynamics of brain activity unfolding over longer temporal scales comprising words, sentences, and larger-scale narrative structures (e.g. Lerner et al., 2011; Chien & Honey, 2020; Chang et al., 2022).

We have clarified how the number of TRs was determined in the revised text:

The number of TRs in our analyses was determined by the duration of the naturalistic spoken story stimuli; e.g. the “I Knew You Were Black” story told live by Carol Daniel for the Moth Radio Hour (<https://themoth.org/stories/i-knew-you-were-black>) was 800 seconds (13 minutes, 20 seconds) long, corresponding to 534 TRs with a 1.5-second TR.

5. In Figure S17, 500 TRs are plotted. Here you can clearly see the sparsity differences between dependencies and transformations. This should be carefully explored and controlled for given the signal processing consequences. Second, how and why a transformer head best picks out a dependency type when its time series is so different from that of the dependency begs for a theory of how an attention head is (if it is) representing information about the dependency, and what that information is.

In Fig. S17, we plot the one-dimensional transformation that best predicts a given dependency for the full 534 TRs of the previously mentioned spoken story “I Knew You Were Black”. For many of these dependencies, we observe correlations between the one-dimensional transformation and the actual dependency on the scale of $r = .25$ to $r = .75$. The goal of this visualization was to provide an intuition for how these transformations behave; that is, to demonstrate that they capture *something* about a given dependency, but—as the reviewer suggests—they are qualitatively different. We expect these to be qualitatively different because the dependencies are binary labels whereas the one-dimensional transformation is a linear combination of continuous-valued activations fluctuating over time. If the reviewer has any suggestions as to how we might better explore the relationship between these one-dimensional transformations and the dependencies, we’re happy to follow up.

How and to what extent transformers encode syntactic structure in attention heads (Raganato et al., 2018; Clark et al., 2019; Jawahar et al., 2019; Abnar & Zuidema, 2020; cf. Htut et al., 2019) and embeddings (Hewitt & Manning, 2019; Liu et al., 2019; Tenney et al., 2019; Ettinger, 2020; Merullo et al., 2023) is an area of active, ongoing research in machine learning and natural language

processing. In the revised manuscript, we’ve added a new Table S2 reporting the dependency prediction scores for the 64-dimensional transformations at the top-performing attention heads. The mean balanced accuracy across top-performing heads is ~0.80, evaluated using out-of-sample prediction, and most dependencies are best predicted by a different head. These results corroborate previous findings in the literature (e.g. Clark et al., 2019) and provide empirical support for the claim that the transformations implemented by individual attention heads “represent” information about certain dependencies (based on a common usage of “represent” from e.g. Haxby et al., 2001).

In line with prior work (Clark et al., 2019; Tenney et al., 2019), we empirically confirmed that the transformations at certain attention heads preferentially encode certain linguistic dependencies in our stimuli (Table S2).

Dependency	Mean performance across all heads	Top-performing head	Performance of top head
prep	0.654	1-11	0.846 (z = 3.0)
pobj	0.631	1-4	0.764 (z = 3.1)
det	0.658	3-1	0.852 (z = 3.2)
nsubj	0.682	3-1	0.849 (z = 2.6)
amod	0.613	3-10	0.745 (z = 3.0)
dobj	0.598	6-11	0.694 (z = 2.5)
advmod	0.622	1-11	0.728 (z = 2.6)
aux	0.686	2-2	0.863 (z = 2.6)
poss	0.646	6-1	0.844 (z = 2.5)
ccomp	0.620	5-11	0.770 (z = 3.4)
mark	0.597	7-10	0.740 (z = 3.0)
prt	0.619	3-1	0.841 (z = 3.1)

Table S2. Dependency prediction scores for transformations at top-performing attention heads. For each linguistic dependency, we used logistic regression with five-fold cross-validation to predict the binary occurrence of the dependency across TRs from the 64-dimensional transformation vectors separately at each attention head (see “Decoding dependency relations” in the Methods section). Given that some dependencies only occur at relatively few TRs (i.e. unbalanced class frequencies), decoding performance was evaluated using balanced classification accuracy (theoretical chance = 0.5). We first report the mean balanced accuracy across all 144 attention heads (mean accuracy = 0.636 across all dependencies). For each dependency, we then report the head with the highest decoding performance (e.g. “1-11” indicates head 11 in layer 1 of BERT). For the top-performing head, we report the balanced accuracy and the z-score for this accuracy relative to the distribution of accuracies across all heads.

References

- Abnar, S., & Zuidema, W. (2020). Quantifying attention flow in transformers. In Jurafsky, D., Chai, J., Schluter, N., & Tetreault, J. (Eds.) *Proceedings of the 58th Annual Meeting of the Association for Computational Linguistics* (pp. 4190–4197). <https://doi.org/10.18653/v1/2020.acl-main.385>
- Aly, M., Chen, J., Turk-Browne, N. B., & Hasson, U. (2018). Learning naturalistic temporal structure in the posterior medial network. *Journal of Cognitive Neuroscience*, *30*(9), 1345–1365. https://doi.org/10.1162/jocn_a_01308
- Antonello, R., & Huth, A. (2023). Predictive coding or just feature discovery? An alternative account of why language models fit brain data. *Neurobiology of Language*. https://doi.org/10.1162/nol_a_00087
- Brennan, J. (2016). Naturalistic sentence comprehension in the brain. *Language and Linguistics Compass*, *10*(7), 299–313. <https://doi.org/10.1111/lnc3.12198>
- Caucheteux, C., Gramfort, A., & King, J. R. (2021). Disentangling syntax and semantics in the brain with deep networks. In Meila, M. & Zhang, T. (Eds.) *International Conference on Machine Learning* (pp. 1336–1348). <https://proceedings.mlr.press/v139/caucheteux21a.html>
- Caucheteux, C., Gramfort, A., & King, J. R. (2022). Deep language algorithms predict semantic comprehension from brain activity. *Scientific Reports*, *12*, 16327. <https://doi.org/10.1038/s41598-022-20460-9>
- Chang, C. H., Nastase, S. A., & Hasson, U. (2022). Information flow across the cortical timescale hierarchy during narrative construction. *Proceedings of the National Academy of Sciences*, *119*(51), e2209307119. <https://doi.org/10.1073/pnas.2209307119>
- Chien, H. Y. S., & Honey, C. J. (2020). Constructing and forgetting temporal context in the human cerebral cortex. *Neuron*, *106*(4), 675–686. <https://doi.org/10.1016/j.neuron.2020.02.013>
- Défossez, A., Caucheteux, C., Rapin, J., Kabeli, O., & King, J. R. (2023). Decoding speech perception from non-invasive brain recordings. *Nature Machine Intelligence*, *5*, 1097–1107. <https://doi.org/10.1038/s42256-023-00714-5>
- Devlin, J., Chang, M. W., Lee, K., & Toutanova, K. (2019, June). BERT: pre-training of deep bidirectional transformers for language understanding. In Burstein, J., Doran, C., & Solorio, T. (Eds.) *Proceedings of the 2019 Conference of the North American Chapter of the Association for Computational Linguistics: Human Language Technologies, Volume 1 (Long and Short Papers)* (pp. 4171–4186). <https://doi.org/10.18653/v1/N19-1423>
- Elhage, N., Nanda, N., Olsson, C., Henighan, T., Joseph, N., Mann, B., Askell, A., Bai, Y., Chen, A., Conerly, T., DasSarma, N., Drain, D., Ganguli, D., Hatfield-Dodds, Z., Hernandez, D., Jones, A., Kernion, J., Lovitt, L., Ndousse, K., Amodei, D., Brown, T., Clark, J., Kaplan, J., McCandlish, S., &

- Olah, C. (2021). A Mathematical Framework for Transformer Circuits. Transformer Circuits Thread. <https://transformer-circuits.pub/2021/framework/index.html>
- Ettinger, A. (2020). What BERT is not: lessons from a new suite of psycholinguistic diagnostics for language models. *Transactions of the Association for Computational Linguistics*, 8, 34–48. https://doi.org/10.1162/tacl_a_00298
- Fedorenko, E., Blank, I. A., Siegelman, M., & Mineroff, Z. (2020). Lack of selectivity for syntax relative to word meanings throughout the language network. *Cognition*, 203, 104348. <https://doi.org/10.1016/j.cognition.2020.104348>
- Fedorenko, E., Hsieh, P. J., Nieto-Castañón, A., Whitfield-Gabrieli, S., & Kanwisher, N. (2010). New method for fMRI investigations of language: defining ROIs functionally in individual subjects. *Journal of Neurophysiology*, 104(2), 1177–1194. <https://doi.org/10.1152/jn.00032.2010>
- Goldstein, A., Wang, H., Niekerken, L., Zada, Z., Aubrey, B., Sheffer, T., Nastase, S. A., Gazula, H., Schain, M., Singh, A., Rao, A., Choe, G., Kim, C., Doyle, W., Friedman, D., Devore, S., Dugan, P., Hassidim, A., Brenner, M., Matias, Y., Devinsky, O., Flinker, A., & Hasson, U. (2023). Deep speech-to-text models capture the neural basis of spontaneous speech in everyday conversations. *bioRxiv*. <https://doi.org/10.1101/2023.06.26.546557>
- Guest, O., & Martin, A. E. (2023). On logical inference over brains, behaviour, and artificial neural networks. *Computational Brain & Behavior*, 6, 213–227. <https://doi.org/10.1007/s42113-022-00166-x>
- Hale, J. T., Campanelli, L., Li, J., Bhattasali, S., Pallier, C., & Brennan, J. R. (2022). Neurocomputational models of language processing. *Annual Review of Linguistics*, 8, 427–446. <https://doi.org/10.1146/annurev-linguistics-051421-020803>
- Hasson, U., Nastase, S. A., & Goldstein, A. (2020). Direct fit to nature: an evolutionary perspective on biological and artificial neural networks. *Neuron*, 105(3), 416–434. <https://doi.org/10.1016/j.neuron.2019.12.002>
- Haxby, J. V., Gobbini, M. I., Furey, M. L., Ishai, A., Schouten, J. L., & Pietrini, P. (2001). Distributed and overlapping representations of faces and objects in ventral temporal cortex. *Science*, 293(5539), 2425–2430. <https://doi.org/10.1126/science.1063736>
- Hewitt, J., & Manning, C. D. (2019). A structural probe for finding syntax in word representations. In Burstein, J., Doran, C., & Solorio, T. (Eds.) *Proceedings of the 2019 Conference of the North American Chapter of the Association for Computational Linguistics: Human Language Technologies, Volume 1 (Long and Short Papers)* (pp. 4129–4138). <https://doi.org/10.18653/v1/N19-1419>
- Htut, P. M., Phang, J., Bordia, S., & Bowman, S. R. (2019). Do attention heads in BERT track syntactic dependencies? *arXiv*. <https://doi.org/10.48550/arXiv.1911.12246>

- Jain, S., Vo, V. A., Wehbe, L., & Huth, A. G. (2023). Computational language modeling and the promise of *in silico* experimentation. *Neurobiology of Language*, 1–27. https://doi.org/10.1162/nol_a_00101
- Jawahar, G., Sagot, B., & Seddah, D. (2019, July). What does BERT learn about the structure of language? In Korhonen, A., Traum, D., & Màrquez, L. (Eds.) *Proceedings of the 57th Annual Meeting of the Association for Computational Linguistics* (pp. 3651–3657). <https://doi.org/10.18653/v1/P19-1356>
- LeBel, A., Wagner, L., Jain, S., Adhikari-Desai, A., Gupta, B., Morgenthal, A., Tang, J., Xu, L., & Huth, A. G. (2023). A natural language fMRI dataset for voxelwise encoding models. *Scientific Data*, 10(1), 555. <https://doi.org/10.1038/s41597-023-02437-z>
- Lerner, Y., Honey, C. J., Silbert, L. J., & Hasson, U. (2011). Topographic mapping of a hierarchy of temporal receptive windows using a narrated story. *Journal of Neuroscience*, 31(8), 2906–2915. <https://doi.org/10.1523/JNEUROSCI.3684-10.2011>
- Liu, N. F., Gardner, M., Belinkov, Y., Peters, M. E., & Smith, N. A. (2019). Linguistic knowledge and transferability of contextual representations. In Burstein, J., Doran, C., & Solorio, T. (Eds.) *Proceedings of the 2019 Conference of the North American Chapter of the Association for Computational Linguistics: Human Language Technologies, Volume 1 (Long and Short Papers)* (pp. 1073–1094). <https://doi.org/10.18653/v1/N19-1112>
- Margulies, D. S., Ghosh, S. S., Goulas, A., Falkiewicz, M., Huntenburg, J. M., Langs, G., Bezgin, G., Eickhoff, S. B., Castellanos, F. X. & Smallwood, J. (2016). Situating the default-mode network along a principal gradient of macroscale cortical organization. *Proceedings of the National Academy of Sciences*, 113(44), 12574–12579. <https://doi.org/10.1073/pnas.1608282113>
- Michelmann, S., Price, A. R., Aubrey, B., Strauss, C. K., Doyle, W. K., Friedman, D., Dugan, P. C., Devinsky, O., Devore, S., Flinker, A., Hasson, U. & Norman, K. A. (2021). Moment-by-moment tracking of naturalistic learning and its underlying hippocampo-cortical interactions. *Nature Communications*, 12, 5394. <https://doi.org/10.1038/s41467-021-25376-y>
- Mikolov, T., Sutskever, I., Chen, K., Corrado, G. S., & Dean, J. (2013). Distributed representations of words and phrases and their compositionality. *Advances in Neural Information Processing Systems*, 26. <https://proceedings.neurips.cc/paper/2013/hash/9aa42b31882ec039965f3c4923ce901b-Abstract.html>
- Naselaris, T., Kay, K. N., Nishimoto, S., & Gallant, J. L. (2011). Encoding and decoding in fMRI. *NeuroImage*, 56(2), 400–410. <https://doi.org/10.1016/j.neuroimage.2010.07.073>
- Nastase, S. A., Liu, Y. F., Hillman, H., Norman, K. A., & Hasson, U. (2020). Leveraging shared connectivity to aggregate heterogeneous datasets into a common response space. *NeuroImage*, 217, 116865. <https://doi.org/10.1016/j.neuroimage.2020.116865>

- Piantadosi, S. (2023). Modern language models refute Chomsky's approach to language. *LingBuzz*.
<https://lingbuzz.net/lingbuzz/007180>
- Radford, A., Kim, J. W., Xu, T., Brockman, G., McLeavey, C., & Sutskever, I. (2023). Robust speech recognition via large-scale weak supervision. In *Proceedings of the 40th International Conference on Machine Learning* (pp. 28492–28518). PMLR.
<https://proceedings.mlr.press/v202/radford23a.html>
- Radford, A., Wu, J., Child, R., Luan, D., Amodei, D., & Sutskever, I. (2019). Language models are unsupervised multitask learners. *OpenAI Blog*.
https://cdn.openai.com/better-language-models/language_models_are_unsupervised_multitask_learners.pdf
- Raganato, A., & Tiedemann, J. (2018). An analysis of encoder representations in transformer-based machine translation. In Linzen, T., Chrupala, G., & Alishahi, A. (Eds.) *Proceedings of the 2018 EMNLP Workshop BlackboxNLP: Analyzing and Interpreting Neural Networks for NLP* (pp. 287–297). <https://doi.org/10.18653/v1/W18-5431>
- Richards, B. A., Lillicrap, T. P., Beaudoin, P., Bengio, Y., Bogacz, R., Christensen, A., Clopath, C., Costa, R. P., de Berker, A., Ganguli, S., Gillon, C. J., Hafner, D., Kepecs, A., Kriegeskorte, N., Latham, P., Lindsay, G. W., Miller, K. D., Naud, R., Pack, C. C., ... Kording, K. P. (2019). A deep learning framework for neuroscience. *Nature Neuroscience*, 22(11), 1761–1770.
<https://doi.org/10.1038/s41593-019-0520-2>
- Samara, A., Eilbott, J., Margulies, D. S., Xu, T., & Vanderwal, T. (2023). Cortical gradients during naturalistic processing are hierarchical and modality-specific. *NeuroImage*, 271, 120023.
<https://doi.org/10.1016/j.neuroimage.2023.120023>
- Schaeffer, R., Khona, M., & Fiete, I. (2022). No free lunch from deep learning in neuroscience: a case study through models of the entorhinal-hippocampal circuit. *Advances in Neural Information Processing Systems*, 35,
https://proceedings.neurips.cc/paper_files/paper/2022/hash/66808849a9f5d8e2d00dbdc844de6333-Abstract-Conference.html
- Simony, E., Honey, C. J., Chen, J., Lositsky, O., Yeshurun, Y., Wiesel, A., & Hasson, U. (2016). Dynamic reconfiguration of the default mode network during narrative comprehension. *Nature Communications*, 7, 12141. <https://doi.org/10.1038/ncomms12141>
- Toneva, M., & Wehbe, L. (2019). Interpreting and improving natural-language processing (in machines) with natural language-processing (in the brain). *Advances in Neural Information Processing Systems*, 32.
<https://proceedings.neurips.cc/paper/2019/hash/749a8e6c231831ef7756db230b4359c8-Abstract.html>

- Vaidya, A. R., Jain, S., & Huth, A. (2022). Self-supervised models of audio effectively explain human cortical responses to speech. In Chaudhuri, K., Jegelka, S., Song, L., Szepesvari, C., Niu, G., & Sabato, S. (Eds.) *International Conference on Machine Learning* (pp. 21927-21944). PMLR. <https://proceedings.mlr.press/v162/vaidya22a.html>
- Van Uden, C. E., Nastase, S. A., Connolly, A. C., Feilong, M., Hansen, I., Gobbini, M. I., & Haxby, J. V. (2018). Modeling semantic encoding in a common neural representational space. *Frontiers in Neuroscience*, 12, 437. <https://doi.org/10.3389/fnins.2018.00437>
- Vo, V. A., Jain, S., Beckage, N., Chien, H. Y. S., Obinwa, C., & Huth, A. G. (2023). A unifying computational account of temporal context effects in language across the human cortex. *bioRxiv*. <https://doi.org/10.1101/2023.08.03.551886>
- Wang, A., Pruksachatkun, Y., Nangia, N., Singh, A., Michael, J., Hill, F., Levy, O. & Bowman, S. (2019). SuperGLUE: a stickier benchmark for general-purpose language understanding systems. *Advances in Neural Information Processing Systems*, 32. https://proceedings.neurips.cc/paper_files/paper/2019/hash/4496bf24afe7fab6f046bf4923da8de6-Abstract.html
- Wang, A., Singh, A., Michael, J., Hill, F., Levy, O., & Bowman, S. (2018, November). GLUE: a multi-task benchmark and analysis platform for natural language understanding. In Linzen, T., Chrupała, G., & Alishahi, A. (Eds.) *Proceedings of the 2018 EMNLP Workshop BlackboxNLP: Analyzing and Interpreting Neural Networks for NLP* (pp. 353–355). <https://doi.org/10.18653/v1/W18-5446>
- Yamins, D. L., & DiCarlo, J. J. (2016). Using goal-driven deep learning models to understand sensory cortex. *Nature Neuroscience*, 19(3), 356–365. <https://doi.org/10.1038/nn.4244>

Reviewer #1 (Remarks to the Author):

The authors have satisfactorily addressed all my concerns and I do not have further comments. I appreciate the authors' remarkable efforts.

Reviewer #1 (Remarks on code availability):

The code is well organized, and I believe it is a usable resource for the community.

Reviewer #2 (Remarks to the Author):

I believe that the reviewers have satisfactorily addressed my comments (thank you!) and made necessary changes to the manuscript. I support this manuscript being published.